# EMPOWERING CHANNEL-OF-MOBILE-EXPERTS WITH INFORMATIVE HYBRID-CAPABILITIES REASONING

## ABSTRACT

Mobile Agents can autonomously execute user instructions, which requires hybrid-capabilities reasoning, including screen summary, subtask planning, action decision and action function. However, existing agents struggle to achieve both decoupled enhancement and balanced integration of these capabilities. While Mixture-of-Experts (MoE) supports capability decoupling, the input-oriented activation prevents the selection of expert aligning with the reasoning stage. To address these challenges, we propose Channel-of-Mobile-Experts (**CoME**), a novel agent architecture consisting of four distinct experts, each aligned with a specific reasoning stage, CoME activates the corresponding expert to generate output tokens in each reasoning stage via *output-oriented activation*. To empower CoME with hybrid-capabilities reasoning, we introduce a progressive training strategy: **Expert-FT** enables decoupling and enhancement of different experts' capability; **Router-FT** aligns expert activation with the different reasoning stage; **CoT-FT** facilitates seamless collaboration and balanced optimization across multiple capabilities. To mitigate error propagation in hybrid-capabilities reasoning, we propose InfoGain-Driven DPO (**Info-DPO**), which uses information gain to evaluate the contribution of each intermediate step, thereby guiding CoME toward more informative reasoning. Comprehensive experiments show that CoME outperforms dense mobile agents and MoE methods on both AITZ and AMEX datasets.

## 1 INTRODUCTION

Mobile Agents can autonomously execute user instructions and have become a prominent research focus in both academia and industry. Their development is characterized by three major trends: (1) from *API invocation* (Wen et al., 2023a; Deng et al., 2024) to *action simulation* (Li et al., 2020a; Li, 2021), enabling adaptation to more complex environment; (2) from *interactive exploration* (Lee et al., 2023; Wen et al., 2023b) to *supervised finetuning* (Zhang & Zhang, 2024; Cheng et al., 2024), allowing to solve more generalized instructions; (3) from *modular framework* (Wang et al., 2024a; Zhang et al., 2025; Li et al., 2024) to *holistic agent* (Chai et al., 2024; Lin et al., 2024; Gou et al., 2024), simplifying system design and training pipelines. Powered by advanced Multi-modal Large Language Models (MLLMs), mobile agents are undergoing a new paradigm shift from *end-to-end prediction* to *step-by-step reasoning*, which improves the accuracy and robustness of action decision (Wu et al., 2024b; Xu et al., 2024; Qin et al., 2025).

Achieving accurate action decisions remains challenging, even with Chain-of-Thought (CoT) prompting (Wei et al., 2022), as the reasoning process typically requires the agent to: perceive the current screen state, plan the next sub-task, generate high-level action decision and low-level action function (Zhang et al., 2024). This process requires multi-dimensional capabilities, which is referred to as **hybrid-capabilities reasoning** in Figure 1. However existing mobile agents either enhance individual capabilities on task-specific datasets (*e.g.,* screen understanding (Zhang et al., 2021; Wang et al., 2021; Li et al., 2021) or action grounding (Chai et al., 2024; Gou et al., 2024)), often lacking capabilities integration; or pre-train on large-scale dataset (Cheng et al., 2024; Wu et al., 2024b; Qin et al., 2025), which can lead to unbalanced performance of different capabilities. Effective methods for decoupled enhancement and balanced integration of multiple capabilities are still lacking. Although Mixture-of-Experts (MoE) achieves partial capability decoupling by using input-oriented activation that routes different input tokens to different experts (Zhou et al., 2022; Jiang et al., 2024; Lieber et al., 2024; Team, 2024; Dai et al., 2024), the ideal hybrid-capabilities reasoning requires

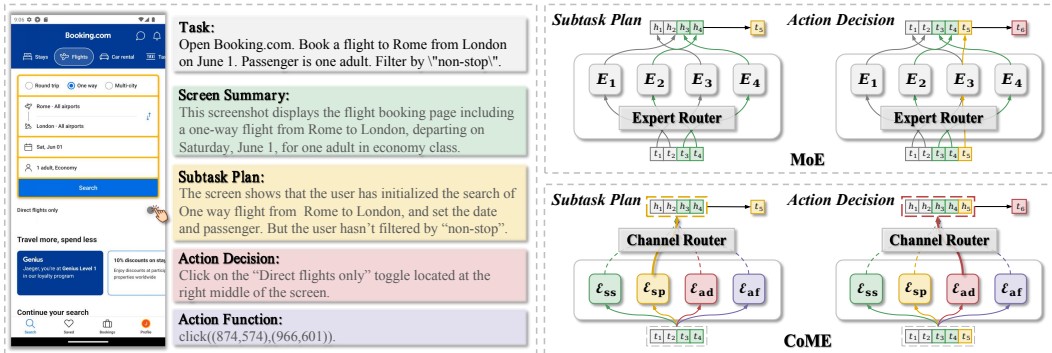

Figure 1: Left is an example of hybrid-capabilities reasoning. Right is the difference between input-oriented activation in MoE and output-oriented activation in CoME.

expert activation to align with the capabilities demanded to generate output tokens at each reasoning stage. Such output-oriented activation, however, is incompatible with the design of MoE.

To address these challenges, we propose a novel agent architecture: **C**hannel-**o**f-**M**obile-**E**xperts (**CoME**), which incorporates four distinct experts to decouple hybrid-capabilities reasoning, each specialized in one of the capabilities: screen summary, subtask plan, action decision, and action function. In contrast to the input-oriented activation in MoE, CoME adopts output-oriented activation as shown in Figure 1. Specifically, CoME forwards the input tokens into each expert and selects the hidden states from the expert aligned with the current reasoning stage to generate each output token. To facilitate CoME with hybrid-capabilities reasoning, we design a progressive training strategy: (1) **Expert-FT** trains FFN layers on capability-specific data to initialize each expert, achieving effective decoupling and enhancement of different capabilities; (2) **Router-FT** trains channel router using expert label of each output token to align expert activation with reasoning stage; (3) **CoT-FT** trains CoME with the hybrid-capabilities reasoning data to enable seamless collaboration and balanced optimization among different experts. Through this progressive curriculum, CoME achieves hybrid-capabilities reasoning by activating specific expert aligned with reasoning stages.

As hybrid-capabilities reasoning spans multiple stages, even minor errors in intermediate steps can propagate and compromise the final outcome. To address this, we propose **InfoGain-Driven DPO (Info-DPO)**, which leverages information gain to quantify the contribution of each intermediate step to the final action prediction. Specifically, we use reward model to estimate the information entropy to the ground truth action before and after each reasoning stage, and use the reduction in entropy as InfoGain reward. Combined with action accuracy reward, we distinguish effective reasoning trajectories to construct high-quality DPO data. Through this mechanism, Info-DPO encourages the model to reinforce informative and reliable intermediate steps, while suppressing those contain distraction reasoning steps. Consequently, the model improves the reasoning accuracy at each stage and mitigates the error propagation throughout the reasoning trajectory. Experiments on the AITZ and AMEX datasets demonstrate that CoME outperforms dense mobile agents (+1.73%) and sparse MoE models (+5.72%) with equivalent activated parameters. Further analysis verifies that CoME achieves accurate expert activation aligned with reasoning stage, while Info-DPO improves the effectiveness of intermediate reasoning steps. Overall, our main contributions can be summarized as follows:

• We propose Channel-of-Mobile-Experts (**CoME**), a novel agent architecture incorporates experts specialized in: screen summary, subtask plan, action decision and action function. CoME employs output-oriented activation to activate appropriate expert at each stage of hybrid-capabilities reasoning.

• We develop a progressive training strategy to empower CoME for hybrid-capabilities reasoning: **Expert-FT** enables decoupling and enhancement of different capabilities to profile specific experts; **Router-FT** aligns expert activation with the different reasoning stage; **CoT-FT** facilitates seamless collaboration and balanced optimization among different experts.

• We introduce InfoGain-Driven DPO (**Info-DPO**), which measures the contribution of intermediate reasoning steps via information gain to suppress invalid reasoning and mitigate error propagation in DPO training. Comprehensive experiments demonstrate the effectiveness CoME.

## 2 RELATED WORKS

### 2.1 AUTONOMOUS MOBILE AGENTS

Mobile Agents can autonomously execute user instruction through API invocation or action simulation (Wen et al., 2023a; Deng et al., 2024; Li et al., 2020a; Li, 2021), which have become a pivotal research spotlight. To equip the agents with mobile knowledge, previous approaches introduce a prior exploration stage (Lee et al., 2023; Wen et al., 2023b; Li et al., 2024), or design multiple specific tasks, such as screen understand (Zhang et al., 2021; Wang et al., 2021; Li et al., 2021), widgets recognition (Chen et al., 2022; Li et al., 2020b; Zheng et al., 2024), GUI transition (Gou et al., 2024; Wu et al., 2024a) and element/action grounding (Cheng et al., 2024; Chai et al., 2024; Bai et al., 2021; Baechler et al., 2024; Qian et al., 2024). Empowered by the advanced MLLMs, some research design some framework consists of multiple stages or agents (Wang et al., 2024a; Li et al., 2024; Wang et al., 2025; Liu et al., 2025). Recently, more works pre-train or finetune on large scale of mixed data to build general mobile agents (Zhang & Zhang, 2024; Wu et al., 2024b; Xu et al., 2024; Qin et al., 2025; Zhang et al., 2024). While existing mobile agents still struggle in capability disentanglement and balanced integration. Therefore, we propose Channel-of-Mobile-Experts (CoME) to facilitate hybrid-capabilities reasoning.

### 2.2 MIXTURE-OF-EXPERTS

Mixture-of-Experts (MoE) (Jacobs et al., 1991) integrates multiple experts and dynamically routes each input to the most relevant expert, to address diverse tasks (Shazeer et al., 2017; Lepikhin et al., 2021; Fedus et al., 2022). Recent works have integrated MoE into LLMs (Jiang et al., 2024; Lieber et al., 2024; Team, 2024; Dai et al., 2024) and MLLMs (Li et al., 2025; Deitke et al., 2024; Wu et al., 2024c) to boost capacity and efficiency by extending FFN layers into multiple experts and activating only top-K experts for each input token. AriaUI (Yang et al., 2024), the first MoE GUI agent, demonstrates the potential of MoE for mobile automation. However, MoE relies on input-oriented activation, which is not optimal for hybrid-capabilities reasoning. In contrast, CoME employs output-oriented activation to align expert activation with the reasoning stage.

## 3 METHODOLOGY

### 3.1 TASK FORMULATION

Given a user instruction $I$, at each step, the **Mobile Agent** $\mathcal{M}$ performs **hybrid-capabilities reasoning** on the current screen state $S$ and action history $H$ to generate the next action $a$. The reasoning trajectory $T$ comprises four distinct stages: screen summary ($T_{\text{ss}}$), subtask planning ($T_{\text{sp}}$), action decision ($T_{\text{ad}}$), and action function invocation ($T_{\text{af}}$). The action $a$ is extracted from $T_{\text{af}}$.

$$a, T = \mathcal{M}(I, S, H), \quad \text{where } T = [T_{\text{ss}}, T_{\text{sp}}, T_{\text{ad}}, T_{\text{af}}]. \tag{1}$$

### 3.2 CHANNEL-OF-MOBILE-EXPERTS (COME)

Mobile agents can automate user instructions through careful hybrid-capabilities reasoning, but still face two main challenges: dense models struggle to achieve decoupled enhancement and balanced integration of different capabilities, while MoE models fail to activate the expert aligned with the reasoning stage.

To address these challenges, we propose Channel-of-Mobile-Experts (**CoME**), a novel agent architecture with *output-oriented activation*. CoME extends FFN module in each layer with four different experts $\left[\mathcal{E}_{\text{ss}}, \mathcal{E}_{\text{sp}}, \mathcal{E}_{\text{ad}}, \mathcal{E}_{\text{af}}\right]$, while shares the same Self-Attn module among experts. In contrast to the input-oriented activation in MoE[1], CoME adopts output-oriented activation to activates expert with the capability required by the current

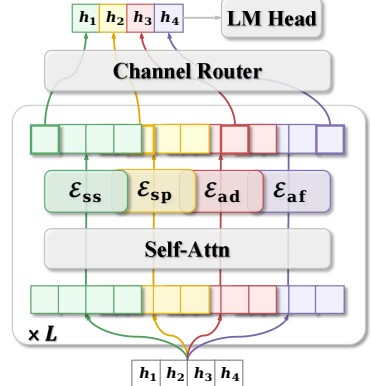

Figure 2: CoME architecture.

---

[1]Limitation of MoE on hybrid-capabilities reasoning is analysed in Appendix A

reasoning stage. Specifically, given the hidden states $\mathbb{H} \in \mathbb{R}^{B \times N \times D}$ after the embedding layer, where $B$, $N$, $D$ stands for the batch size, sequence length and hidden size. CoME first repeats the hidden states for $E$ times $\mathbb{H} \to \widetilde{\mathbb{H}} \in \mathbb{R}^{B \times N \times E \times D}$, where $E$ represents the number of experts and $\widetilde{\mathbb{H}}$ represents the channel hidden states. At the $l$-th layer in CoME:

$$\begin{aligned}
\widetilde{\mathbb{H}}^{(l)} &= \text{Self-Attn}\left(\widetilde{\mathbb{H}}^{(l-1)}\right) \\
\widetilde{\mathbb{H}}_{[e]}^{(l)} &= \text{FFN}_e\left(\widetilde{\mathbb{H}}_{[e]}^{(l)}\right), \quad e \in [1, \cdots, E]
\end{aligned} \tag{2}$$

After obtaining the channel hidden state $\widetilde{\mathbb{H}}^{(L)}$ for the last layer, the most important step in CoME to achieve output-oriented activation is to select the hidden state from the corresponding expert channel according to the current reasoning stage. The channel router $\mathbf{W}_c \in \mathbb{R}^{ED \times E}$ will project the flattened hidden states into $E$-dimension channel logits $\mathbb{C} \in \mathbb{R}^{B \times N \times E}$, which will be used to fuse the hidden states from different expert channels to generate the final hidden states $\widehat{\mathbb{H}} \in \mathbb{R}^{B \times N \times D}$, thus achieving output-oriented activation. Then the final hidden states will then be forwarded into LM-Head.

$$\begin{aligned}
\mathbb{C} &= \text{flatten}\left(\widetilde{\mathbb{H}}^{(L)}\right) \times \mathbf{W}_c, \\
\widehat{\mathbb{H}}_{[b,n]} &= \sum_{e=1}^{E} \text{softmax}(\mathbb{C})_{[e]} \cdot \widetilde{\mathbb{H}}_{[e]}^{(L)},
\end{aligned} \tag{3}$$

### 3.3 PROGRESSIVE TRAINING STRATEGY

In order to empower CoME with hybrid-capabilities reasoning, we introduce a progressive training strategy: (1) **Expert Finetuning (Expert-FT)**, which explicitly decouples and enhances different capabilities; (2) **Router Finetuning (Router-FT)**, which allows expert activation aligned with the current reasoning stage; (3) **Chain-of-Thought Finetuning (CoT-FT)**, which facilitates seamless collaboration and balanced optimization among experts. Details in the following sections.

#### 3.3.1 STAGE-1: EXPERT FINETUNING (EXPERT-FT)

The hybrid-capabilities reasoning can be divided into four stages: screen summary, subtask plan, action decision and action function (Zhang et al., 2024), which is explicitly decoupled with different experts in CoME. To initialize and enhance these specialized experts, we train the FFN modules in Qwen2VL (Wang et al., 2024b) dense model on the dataset $\mathcal{D}_e$ of a specific ability respectively.

$$\mathcal{L}_{\text{Expert-FT}} = -\mathbb{E}_{\{x,y\} \sim \mathcal{D}_e}\left[\log \pi_{\mathcal{M}_e}(y \mid x)\right], \quad \text{where } e \in \left[\mathcal{E}_{\text{ss}}, \mathcal{E}_{\text{sp}}, \mathcal{E}_{\text{ad}}, \mathcal{E}_{\text{af}}\right] \tag{4}$$

We extract the FFN layers in the specialized experts $\left[\mathcal{E}_{\text{ss}}, \mathcal{E}_{\text{sp}}, \mathcal{E}_{\text{ad}}, \mathcal{E}_{\text{af}}\right]$ to assemble CoME.

$$\text{FFN}_{\text{CoME}}^{(l)} = \left[\text{FFN}_{\text{ss}}^{(l)}, \text{FFN}_{\text{sp}}^{(l)}, \text{FFN}_{\text{ad}}^{(l)}, \text{FFN}_{\text{af}}^{(l)}\right]$$

#### 3.3.2 STAGE-2: ROUTER FINETUNING (ROUTER-FT)

After the Expert-FT, CoME has potentially mastered the capabilities required by hybrid-capabilities reasoning, thus it is necessary to enable output-oriented activation in CoME to align expert activation with the reasoning stage. We augment the hybrid-capabilities reasoning data $\mathcal{D}$ with the activated expert label of each output token according to the reasoning stage it is in. During training, we optimize the channel router using the Cross-Entropy loss $\mathcal{L}_{\text{R-CE}}$ between the predicted channel logits $\mathbb{C} \in \mathbb{R}^{B \times N \times E}$ and the expert labels $\mathcal{C} \in \mathbb{R}^{B \times N}$. To further supervise expert activation, we apply a Router Norm loss $\mathcal{L}_{\text{R-Norm}}$ as a regularization term to suppress irrelevant experts.

$$\begin{aligned}
\mathcal{L}_{\text{R-CE}} &= -\mathbb{E}_{\{\mathcal{C}, \mathbb{C}\} \sim \mathcal{D}}\left[\mathcal{C} \cdot \log \text{softmax}(\mathbb{C})\right] \\
\mathcal{L}_{\text{R-Norm}} &= -\mathbb{E}_{\{\mathcal{C}, \mathbb{C}\} \sim \mathcal{D}}\left[\|\text{softmax}(\mathbb{C}), \text{onehot}(\mathcal{C})\|_2^2\right] \\
\mathcal{L}_{\text{Router-FT}} &= \mathcal{L}_{\text{R-CE}} + \mathcal{L}_{\text{R-Norm}}
\end{aligned} \tag{5}$$

### 3.3.3 STAGE-3: CHAIN-OF-THOUGHT FINETUNING (CoT-FT)

CoME achieves the decoupling and enhancement of multi-dimensional capabilities through Expert-FT, and aligns the expert activation with the reasoning stage through Router-FT. To further improve hybrid-capabilities reasoning, we design CoT Finetuning, which aims to simultaneously facilitates seamless collaboration and balanced optimization among different experts. During training, we use SFT loss $\mathcal{L}_{\text{SFT}}$ to optimize the hybrid-capabilities reasoning, and use a regularization term $\mathcal{L}_{\text{R-Norm}}$ to constrain expert activation during reasoning.

$$\mathcal{L}_{\text{SFT}} = -\mathbb{E}_{\{I,S,H,T\}\sim\mathcal{D}}\big[\log \pi_{\mathcal{M}}\big(T \,|\, I, S, H\big)\big]$$
$$\mathcal{L}_{\text{CoT-FT}} = \mathcal{L}_{\text{SFT}} + \gamma \cdot \mathcal{L}_{\text{R-Norm}} \tag{6}$$

### 3.4 INFOGAIN-DRIVEN DPO

Chain-of-Thought stimulates the model's reasoning ability through step-by-step thinking to get more accurate results (Wei et al., 2022). However, the intermediate reasoning steps may contain some errors, which will impact the reasoning accuracy because of the error propagation as the CoT length increases. In addition, reasoning trajectories that reach the correct answer via flawed steps should be suppressed, while those with logical steps but incorrect outcomes may still be beneficial. Therefore the key to improve the accuracy of reasoning is to ensure that each intermediate step has a positive contribution to the final answer. Since hybrid-capabilities reasoning is naturally a multi-stage reasoning structure, we can use the information gain of each intermediate stage to measure its contribution (Ton et al., 2024).

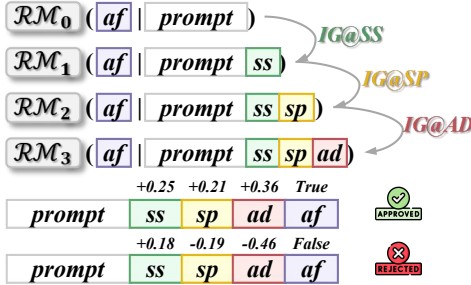

Figure 3: An example of InfoGain reward.

$$IG(T_e) = \log \frac{p\big(T_{\text{af}} \,|\, I, S, H, T_{[0:e]}\big)}{p\big(T_{\text{af}} \,|\, I, S, H, T_{[0:e-1]}\big)}, \quad \text{where } e \in [1, 2, 3] \text{ and } T_{[0:0]} = \phi \tag{7}$$

In order to estimate the information gain of the $e$-th reasoning stage, we train the reward model $\mathcal{RM}_e$, using the reasoning stages $T_{[0:e]}$ as additional inputs to directly predict the final action function $T_{\text{af}}$.

$$\mathcal{L}_{\mathcal{RM}_e} = -\mathbb{E}_{\{I,S,H,T\}\sim\mathcal{D}}\big[\log \pi_{\mathcal{RM}_e}\big(T_{\text{af}} \,|\, I, S, H, T_{[0:e]}\big)\big], \quad \text{where } e \in [0, 1, 2, 3] \tag{8}$$

Therefore, the information gain of the $e$-th reasoning stage in Eq 7 can be approximated by Eq 9. Specifically, we denote the information gain in screen summary, subtask plan and action decision stages as *IG@SS*, *IG@SP* and *IG@AD* respectively as shown in Figure 3. Then we can calculate the reasoning-level InfoGain reward $\mathcal{R}_{\text{IG}} = IG@SS + IG@AP + IG@AD$ and the InfoPos reward $\mathcal{R}_{\text{IG}}^+ = \mathbb{1}_{IG@SS>0} \cdot \mathbb{1}_{IG@AP>0} \cdot \mathbb{1}_{IG@AD>0}$.

$$IG(T_e) \approx \log \frac{\pi_{\mathcal{RM}_e}\big(T_{\text{af}} \,|\, I, S, H, T_{[0:e]}\big)}{\pi_{\mathcal{RM}_{e-1}}\big(T_{\text{af}} \,|\, I, S, H, T_{[0:e-1]}\big)}, \quad \text{where } e \in [1, 2, 3] \text{ and } T_{[0:0]} = \phi \tag{9}$$

Analogous to the outcome-level reward, we define the action-level accuracy reward $\mathcal{R}_{\text{ACC}}$ between the predict action $a$ and ground truth action $\hat{a}$ in Eq 10. For click actions, $\mathcal{R}_{\text{ACC}}$ is computed based on the distance between the predicted and annotated coordinates; for input actions, it is the F1 score between the predicted and reference text; and for other actions, it is a binary exact-match indicator.

$$\mathcal{R}_{\text{ACC}} = \begin{cases} 1 - \max(\text{dis}(a, \hat{a}) \,/\, \delta_d, 1), & \text{click action}, \delta_d \text{ is the distance threshold} \\ \max(\text{f1}(a, \hat{a}), 0) \cdot \mathbb{1}_{\text{f1}>\delta_f}, & \text{input action}, \delta_f \text{ is the F1 score threshold} \\ \mathbb{1}_{a=\hat{a}}, & \text{other actions} \end{cases} \tag{10}$$

When generating DPO data, we sample $K$ reasoning trajectories for each action, and calculate the CoT-reward $\mathcal{R}_{\text{CoT}} = \mathcal{R}_{\text{IG}} \cdot \mathcal{R}_{\text{ACC}}$. We select the trajectory with the highest $\mathcal{R}_{\text{CoT}}$ as well as $\mathcal{R}_{\text{IG}}^+ = 1$ from the trajectories that hit the labeled action as the *chosen output*, which means that all of the

Table 1: **Main results on AITZ dataset of different action.** Results with $*$ are reported from (Zhang et al., 2024). Methods with $\dagger$ are finetuned with hybrid-capabilities reasoning. The best overall result is marked **bold** and the second-best one is marked underline.

| Method | #Params | SCROLL | CLICK | | TYPE | | PRESS | STOP | Overall | |
| --- | --- | --- | --- | --- | --- | --- | --- | --- | --- | --- |
| | | | type | match | type | match | | | type | match |
| Auto-GUI$*$ (Zhang & Zhang, 2024) | 700M | 61.40 | 74.56 | 32.20 | 87.80 | 81.40 | 57.70 | 74.40 | **82.98** | 47.69 |
| Qwen2VL$\dagger$ (Wang et al., 2024b) | 2B | 23.05 | 78.38 | 43.74 | 60.40 | 59.40 | 52.22 | 47.02 | 62.25 | 43.80 |
| ShowUI (Lin et al., 2024) | 2B | 23.79 | 53.29 | 40.34 | 89.60 | 85.20 | 59.00 | 87.55 | 64.22 | 49.55 |
| UITars (Qin et al., 2025) | 2B | 30.17 | 83.81 | 55.25 | 83.97 | 84.77 | 53.78 | 61.39 | 74.23 | 55.63 |
| SeeClick (Cheng et al., 2024) | 7B | 11.14 | 69.92 | 52.96 | 53.80 | 53.00 | 67.88 | 55.36 | 62.93 | 49.11 |
| Qwen2VL$\dagger$ (Wang et al., 2024b) | 7B | 43.28 | 83.97 | 55.30 | 51.80 | 51.40 | 56.92 | 64.87 | 71.59 | 54.46 |
| UGround (Gou et al., 2024) | 7B | 58.22 | 80.94 | 58.48 | 82.56 | 73.85 | 58.22 | 68.78 | 74.54 | 60.19 |
| OS-Atlas (Wu et al., 2024b) | 7B | 76.12 | 75.82 | 54.83 | 89.80 | 88.60 | 68.67 | 81.75 | 77.83 | 65.11 |
| UITars (Qin et al., 2025) | 7B | 56.50 | 84.87 | 63.87 | 85.97 | 85.77 | 58.22 | 71.29 | 78.07 | 65.41 |
| MolmoE$\dagger$ (Deitke et al., 2024) | 1B | 28.19 | 79.84 | 33.17 | 65.20 | 62.00 | 37.07 | 54.96 | 65.96 | 38.23 |
| Qwen2VL-MoE$\dagger$ | 3B | 48.75 | 80.03 | 59.43 | 77.80 | 74.00 | 57.70 | 70.83 | 72.94 | 60.69 |
| AriaUI (Yang et al., 2024) | 3.9B | 53.73 | 85.51 | 60.20 | 84.20 | 80.80 | 63.70 | 76.38 | 78.53 | 63.56 |
| DeepSeekVL2$\dagger$ (Wu et al., 2024c) | 4.5B | 17.94 | 75.35 | 19.98 | 50.90 | 46.69 | 14.36 | 24.25 | 55.36 | 22.55 |
| **CoME** | 5B | 52.07 | 83.83 | 65.22 | 88.20 | 83.80 | 59.53 | 83.33 | 78.60 | **66.98** |

reasoning stages are on the correct direction and lead to the most approximate answer to the labeled action. The *rejected output* is selected from the trajectories that don't hit the ground truth action and has the lowest $\mathcal{R}_{IG}$. More details about the DPO data selection can be found in Appendix B. After obtaining the DPO dataset $\mathcal{D}^*$, we can train CoME using the following DPO loss together with SFT loss $\mathcal{L}_{SFT}$ and Router Norm loss $\mathcal{L}_{R\text{-}Norm}$:

$$\mathcal{L}_{IG\text{-}DPO} = -\mathbb{E}_{\{I,S,H,T^+,T^-\}\sim\mathcal{D}^*}\left[\log\sigma(\beta\log\frac{\pi_\mathcal{M}(T^+|I,S,H)}{\pi_{\mathcal{M}_{ref}}(T^+|I,S,H)} - \beta\log\frac{\pi_\mathcal{M}(T^-|I,S,H)}{\pi_{\mathcal{M}_{ref}}(T^-|I,S,H)})\right] \quad (11)$$
$$+ \alpha\cdot\mathcal{L}_{SFT} + \gamma\cdot\mathcal{L}_{R\text{-}Norm}$$

## 4 EXPERIMENT

### 4.1 EXPERIMENT SETUP

**Datasets.** We train and evaluate CoME on two widely used mobile datasets: (1) **AITZ** (Zhang et al., 2024) contains 2.5k instructions from 70+ apps with chain-of-action-thought. (2) **AMEX** (Chai et al., 2024) contains 104k screenshots and 3k instructions with element groundings and functionality descriptions. We unify the action space: [*click, input, swipe, enter, back, home, finish*].

**Baselines.** We compare CoME with two types of baselines: (1) **Dense mobile agents** pretrained on the mobile datasets, such as Auto-GUI (Zhang & Zhang, 2024), SeeClick (Cheng et al., 2024), SphAgent (Chai et al., 2024), Os-Atlas (Wu et al., 2024b), UGround (Gou et al., 2024), ShowUI (Lin et al., 2024) and UITars (Qin et al., 2025). (2) **Sparse MoE models** pretrained on general multi-modal dataset and finetuned on AITZ/AMEX, including MolmoE (Deitke et al., 2024) and DeepSeekVL2 (Wu et al., 2024c). Specially, AriaUI (Yang et al., 2024) is a native MoE mobile agent. Moreover, we also choose Qwen2VL and Qwen2VL-MoE with hybrid-capabilities reasoning as the basic dense and sparse baselines.

**Metrics.** Following Auto-GUI (Zhang & Zhang, 2024), we use **action type** to access the accuracy of the predicted action type and use **action match** to measure both the type and the parameter.

More details about the datasets, baselines, metrics and implementation are in Appendix D

### 4.2 MAIN RESULTS

We compare **CoME** against 13 baselines on AITZ and AMEX datasets. Following prior works (Zhang et al., 2024; Chai et al., 2024), we report action type and action match metrics.

**AITZ.** As shown in Table 1, CoME achieves the highest overall action match accuracy. Compared to dense mobile agents,CoME yields an improvement of 11.35% over 2B-series methods and 1.57%

Table 2: **Main results on AMEX dataset of different apps**. Results with [*] are reported from (Chai et al., 2024). Methods with [†] are finetuned with hybrid-capabilities reasoning. The best overall result is marked **bold** and the second-best one is marked underline.

| Method | #Params | Gmail | Booking | Music | SHEIN | News | CM | ToDo | Signal | Yelp | Overall |
|---|---|---|---|---|---|---|---|---|---|---|---|
| Qwen2VL[†] (Wang et al., 2024b) | 2B | 37.4 | 35.2 | 40.1 | 33.7 | 50.0 | 47.5 | 41.8 | 56.7 | 40.5 | 38.53 |
| ShowUI (Lin et al., 2024) | 2B | 52.2 | 33.6 | 68.8 | 55.3 | 51.8 | 57.4 | 51.9 | 69.7 | 50.2 | 47.74 |
| UITars (Qin et al., 2025) | 2B | 59.4 | 47.9 | 55.4 | 53.0 | 65.1 | 62.3 | 64.7 | 61.7 | 58.1 | 54.43 |
| SeeClick[*] (Cheng et al., 2024) | 7B | 28.2 | 29.4 | 18.1 | 20.0 | 30.0 | 53.1 | 30.7 | 37.1 | 27.4 | 30.44 |
| Qwen2VL[†] (Wang et al., 2024b) | 7B | 57.6 | 58.4 | 56.5 | 47.3 | 64.2 | 66.3 | 60.9 | 72.8 | 54.8 | 57.99 |
| UGround (Gou et al., 2024) | 7B | 70.9 | 68.8 | 72.7 | 63.7 | 77.7 | 67.7 | 73.7 | 80.1 | 67.6 | 69.12 |
| OS-Atlas (Wu et al., 2024b) | 7B | 74.4 | 69.7 | 74.6 | 64.0 | 80.7 | 63.6 | 65.3 | 83.3 | 62.3 | 69.99 |
| SphAgent[*] (Chai et al., 2024) | 7B | 61.7 | 68.2 | 77.7 | 72.0 | 71.9 | 64.6 | 79.6 | 71.3 | 69.6 | 70.71 |
| UITars (Qin et al., 2025) | 7B | 67.7 | 70.0 | 71.8 | 63.8 | 71.5 | 67.7 | 77.0 | 86.4 | 72.8 | 70.33 |
| MolmoE[†] (Deitke et al., 2024) | 1B | 38.7 | 28.6 | 28.1 | 27.2 | 45.3 | 22.8 | 37.6 | 38.2 | 34.0 | 31.56 |
| Qwen2VL-MoE[†] | 3B | 65.1 | 63.7 | 57.8 | 63.0 | 78.0 | 60.3 | 70.2 | 76.5 | 61.1 | 64.56 |
| AriaUI (Yang et al., 2024) | 3.9B | 63.1 | 62.3 | 68.5 | 58.9 | 83.0 | 54.7 | 62.5 | 83.3 | 66.9 | 64.10 |
| DeepSeekVL2[†] (Wu et al., 2024c) | 4.5B | 43.06 | 36.86 | 52.51 | 42.16 | 42.66 | 51.12 | 46.99 | 61.72 | 38.91 | 42.22 |
| **CoME** | 5B | 76.2 | 72.6 | 81.0 | 64.3 | 81.2 | 63.2 | 72.6 | 78.4 | 66.9 | **72.61** |

over 7B-series methods, using only 5B activated parameters. Moreover, CoME surpasses MoE-based methods by 3.42% on the overall accuracy. Baseline methods generally underperform on the CLICK action, while CoME achieves the highest accuracy of 65.22% (+1.45%), because CLICK is a representative hybrid-capabilities task that is much more challenging. Baseline methods also show imbalanced performance across action types—for example, ShowUI reaches 87.55% on STOP but only 23.79% on SCROLL. In contrast, CoME achieves a higher relative improvement (+11.56%) and lower bias (4.41) in Table 8, indicating more balanced gains across actions.

**AMEX.** As shown in Table 2, CoME achieves the best overall performance (72.61%) across the nine apps, surpassing the dense model (+1.90%) and the sparse MoE (+8.05%). Compared with OS-Atlas, SphAgent and UITars pre-trained with large scale mobile data then finetuned on AMEX without long CoT reasoning, CoME could surpasses them by 2.26% on average, only using AMEX data thought hybrid-capabilities reasoning, which proves that CoME architecture can better activate multi-dimensional capabilities to achieve more effective CoT reasoning. Comparison with basic Qwen2VL (dense) and Qwen2VL-MoE (sparse) baselines demonstrates the CoME achieves more effective capability disentanglement, enhancement, and activation.

### 4.3 ABLATION ANALYSIS

We designed comprehensive ablation experiments to analyze the effects of different training stages and strategies. As shown in Table 3, Info-DPO contributes most to the action prediction (+4.68%), proving that using information gain to distinguish reasoning trajectories can mitigate error propagation and improve reasoning accuracy. Further, removing Router-FT leads to an accuracy decline (-4.08%), indicating that

Table 3: Ablation analysis on AITZ and AMEX.

| Method | AITZ | AMEX | AVERAGE |
|---|---|---|---|
| CoME | 66.98 | 72.61 | 69.78 |
| - w/o Info-DPO | 62.93 | 67.28 | 65.10 |
| - w/o Router-FT | 60.05 | 62.00 | 61.02 |
| - w/o Expert-FT | 57.07 | 64.47 | 60.74 |
| - Info-DPO w/o $\mathcal{L}_{\text{R-Norm}}$ | 65.96 | 70.90 | 68.42 |
| - CoT-FT w/o $\mathcal{L}_{\text{R-Norm}}$ | 62.57 | 64.24 | 63.40 |

Router-FT enables expert activation aligned with the reasoning stage. Moreover, no prior Expert-FT can not fully release the expert's specialized capability (-4.36%). Including router norm $\mathcal{L}_{\text{R-Norm}}$ on expert activation is necessary in CoT-FT (+1.70%) and Info-DPO (+1.36%) as well.

### 4.4 COMPREHENSIVE ANALYSIS OF COME DESIGN

#### 4.4.1 ANALYSIS OF COME ARCHITECTURE.

***Q1: How CoME performs across different capabilities?*** CoME integrates four specialized experts: screen summary ($\mathcal{E}ss$), subtask plan ($\mathcal{E}sp$), action decision ($\mathcal{E}ad$), and action function ($\mathcal{E}af$). We compare CoME with Dense and MoE model, and use specific-capability expert as reference baseline in Figure 4. Dense and MoE method struggle in later reasoning stages, because the first two stages (screen summary and subtask plan) account for 80% of the tokens, which dominate CoT training and

leave later stages inadequately trained. Conversely, by activating the expert aligned with reasoning stage, **CoME enables CoT-FT to optimize each expert in its specialized stage, resulting in balanced improvements across all capabilities.** The action-function expert achieves superior performance because of leveraging ground-truth action decisions as input, whereas others depend on their own predicted decisions. By disentangling and fine-tuning each expert for its specific capability, CoME strengthens intermediate reasoning and effectively mitigates error propagation.

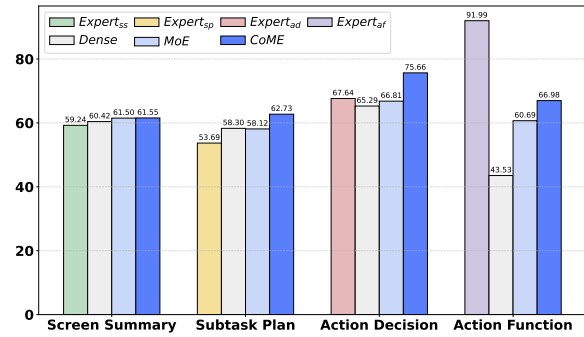

Figure 4: Performance on different reasoning stages.

***Q2: Why CoME improves on hybrid-capabilities reasoning?*** We compare CoME against MoE by analyzing expert activation distributions and selection accuracy at each stage. As shown in Figure 5, expert activation in CoME exhibits clear stage preference (e.g., $\mathcal{E}_{ss}$ for initial screen summary and $\mathcal{E}_{af}$ for final action function) and achieves 99% selection accuracy, whereas MoE activation remain uniformly distributed and poorly aligned with reasoning stages. The experiment demonstrates that **CoME achieves output-oriented activation in hybrid-capabilities reasoning that successfully activate the expert with corresponding capability required by the reasoning stage.**

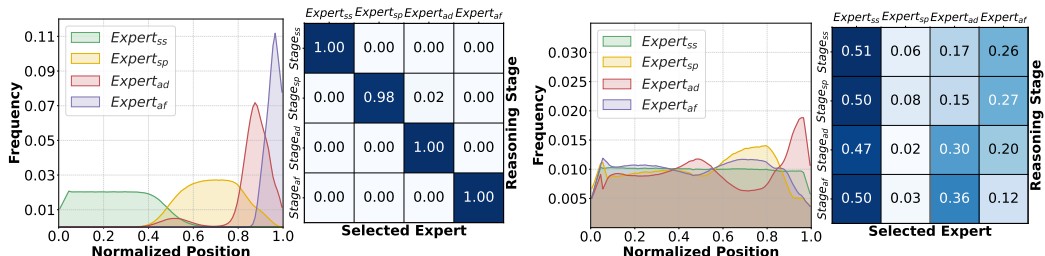

Figure 5: Expert distribution comparison. Left is CoME and right is MoE.

### 4.4.2 ANALYSIS OF INFO-DPO

***Q3: How Info-DPO performs compared with Rule-DPO?*** We compare Info-DPO with Rule-based reward DPO (Rule-DPO) in Table 4. Experimental results shows that Info-DPO surpasses Rule-DPO on both AITZ (+1.61%) and AMEX (+2.41%), and outperforms Rule-DPO across all action categories. For click actions, Rule-DPO yields marginal overall improvements (+0.51% and –0.08%), whereas Info-DPO achieves substantial gains (+3.45% and +1.70%). Because click action is highly sensitive to intermediate error: an incorrect element description can lead to large coordinate deviations. Unlike Rule-DPO with no supervision over intermediate reasoning, **Info-DPO employs information gain to identify and reinforce valid intermediate steps, thereby mitigating error propagation**.

Table 4: Comparison of DPO Strategies.

| Method | AITZ | | | | | | AMEX | | | | | |
|---|---|---|---|---|---|---|---|---|---|---|---|---|
| | Scroll | Click | Type | Press | Stop | Overall | Scroll | Click | Type | Press | Stop | Overall |
| CoT-FT | 51.74 | 61.77 | 70.00 | 63.44 | 68.84 | 62.25 | 70.83 | 66.39 | 79.16 | 70.66 | 51.14 | 67.28 |
| Rule-DPO | 51.98 | 62.26 | **86.20** | **61.80** | 80.36 | 65.37 | 82.34 | 66.31 | 78.38 | 48.00 | 58.35 | 70.20 |
| Info-DPO | **52.07** | **65.22** | 83.80 | 59.53 | **83.33** | **66.98** | 82.43 | 68.09 | 82.58 | 70.67 | 67.74 | 72.61 |

***Q4: How InfoGain-reward performs on reasoning trajectory evaluation?*** Figure 6 presents the Information gain distribution for the sampled data and DPO data. As reasoning progresses, the difference between correct and wrong reasoning increases (0.0766 / 0.0673 → 0.0582 / -0.4252). This confirms that the **InfoGain-reward can evaluate and distinguish the reasoning trajectory, where correct reasoning brings positive InfoGain while wrong reasoning results in negative InfoGain**. Furthermore, filtering DPO data with InfoGain-reward raises the information gain of the selected trajectories, demonstrating its effectiveness in removing invalid intermediate steps.

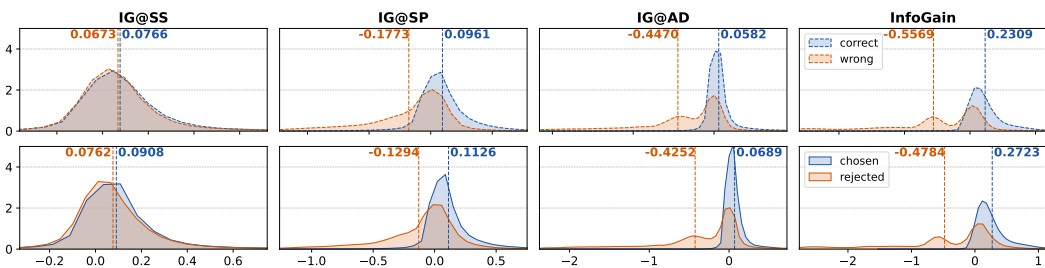

Figure 6: Information gain distribution. Above is the sampled data. Below is the DPO data. We illustrate the information gain at screen summary, subtask plan and action decision stage, as well as the total information gain of the entire reasoning trajectory.

***Q5: How to select effective data for Info-DPO?*** For each action, we sample ten reasoning trajectories and construct DPO pairs using one of three strategies: (1) **cc**: pair two correct trajectories; (2) **cw**: pair a correct trajectory with a wrong one; and (3) **lw**: pair the label with an wrong trajectory. Details in Appendix B. Experimental results in Table 5 and Table 9 show that, higher InfoGain (informative contribution) and InfoPos (intermediate validity) leads to higher action accuracy. While, including labels as chosen outputs (lw) provides no gain because of the distribution gap; introducing suboptimal correct reasoning as rejected outputs (cc) further mitigates invalid intermediate steps. Thus, **the most effective pairing strategy is to pair trajectories with high information gain as chosen outputs and those exhibiting incorrect outcomes or invalid intermediate steps as rejected outputs**.

***Q6: Which reward strategy is better for Info-DPO?*** We analyse the impact of different reward strategies on Info-DPO, including different model scales and calculation methods. As shown in Table 6, using **7B reward model can achieve comparable results of 72B reward model and even better**, because better fitness and lower loss of 72B model results in a smaller InfoGain calculated based on the difference, which reduces the distinguishing ability of the reward model. Moreover, **using continuous reward function of 0-1 is better than the discrete reward function of 0/1**, because more fine-grained evaluation is important for click and input action as shown in Table 10.

<table>
<tr><td colspan="4">Table 5: Analysis of data pairing strategies.</td></tr>
<tr><td>**Strategy**</td><td>**InfoGain**</td><td>**InfoPos**</td><td>**Action Acc**</td></tr>
<tr><td>cc+cw+lw</td><td>-0.503 / 0.264</td><td>0.029 / 0.611</td><td>77.60 / 65.96</td></tr>
<tr><td>cc+cw</td><td>-0.478 / **0.272**</td><td>0.030 / **0.704**</td><td>78.60 / 66.98</td></tr>
<tr><td>cw+lw</td><td>-0.712 / 0.259</td><td>0.038 / 0.463</td><td>76.88 / 64.07</td></tr>
<tr><td>cw</td><td>-0.714 / 0.269</td><td>0.040 / 0.565</td><td>77.75 / 65.93</td></tr>
</table>

Table 6: Analysis of reward strategies.

| Method | InfoGain | InfoPos | Action |
|---|---|---|---|
| Reward-7B | 0.9841 | 0.6610 | 66.98 |
| Reward-72B | 0.8365 | 0.4533 | 66.85 |
| Continuous | 0.9841 | 0.6610 | 66.98 |
| Discrete | 0.7541 | 0.5018 | 65.51 |

### 4.4.3 ANALYSIS OF EFFICIENCY

***Q7: How efficient is CoME?*** We compare CoME with Qwen2VL dense and MoE model in Table 7, **CoME achieves better action accuracy while maintaining lower GPU memory usage**. The memory advantage comes from parameters being the main footprint, with only minor overhead from the KV cache during inference. More details in Appendix I.

Table 7: Efficiency analysis of CoME architecture.

| Method | GPU Mem (GB) | | Accuracy (%) | |
|---|---|---|---|---|
| | train | infer | type | match |
| Qwen2VL-7B | 31.62 | 16.52 | 71.59 | 54.46 |
| Qwen2VL-MoE | 30.70 | 22.32 | 72.94 | 60.69 |
| CoME | 18.52 | 11.69 | 78.60 | 66.98 |

## 5 CONCLUSION

We propose Channel-of-Mobile-Experts (CoME), a novel agent architecture implemented with output-oriented activation, to activate the corresponding expert aligned with different stage in hybrid-capabilities reasoning. We develop a progressive curriculum (Expert-FT, Router-FT, CoT-FT) to achieve decoupled enhancement and balanced integration of different capabilities. To mitigate error propagation in reasoning, we introduce InfoGain-Driven DPO, which uses information gain to reinforce informative intermediate steps. CoME performs best on both AITZ and AMEX, and extensive experiments demonstrate the effectiveness of the novel architecture and training strategies.

## 6 ETHICS STATEMENT

We carefully reviewed our dataset to eliminate any elements that might compromise personal privacy, ensuring the highest level of privacy protection. We recommend that when deploying our model in real-world environments, the entire execution process should be carried out under human supervision to prevent unexpected errors that could lead to the loss of user information or property.

## 7 REPRODUCIBILITY STATEMENT

To ensure the reproducibility of the results, we set the random seed to 1234 and the temperature to 0 during the experiments, so as to eliminate the interference of randomness on the experimental outcomes. We introduce the experimental implementation details in Appendix D and the DPO data pairing methods in Appendix B. We will release all the data and code once the internal open-source review has been completed.

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

## A  LIMITATION OF MIXTURE-OF-EXPERTS ON HYBRID-CAPABILITIES REASONING

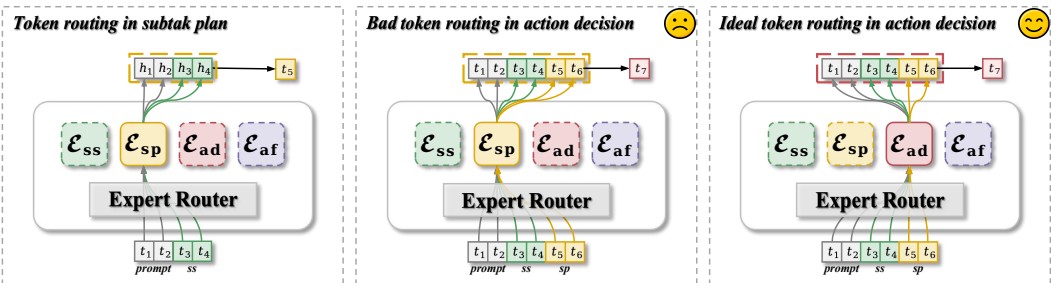

Figure 7: A case to show the limitation of MoE on hybrid-capabilities reasoning.

Mixture-of-Experts (MoE) (Jacobs et al., 1991) architectures have gradually gained traction in LLMs. By introducing multiple FFN modules as experts alongside a routing/gating mechanism, MoE sparsely activates only a subset of the experts for each token in the input sequence, thereby enabling an increase in total parameter while keeping computational costs relatively manageable (Zhou et al., 2022; Jiang et al., 2024; Dai et al., 2024). By assigning different subsets in the input to different experts, MoE implicitly links distinct capabilities to some specific experts, thereby achieving capability disentanglement in hybrid-capability scenarios. However, this represents an input-oriented activation, which is better for handling multi-task or multi-modal inputs.

Mobile Agents present an even more complex scenario. In order to achieve accurate action prediction, the agent need to perceive the current screen state, plan the next sub-task, then generate the high-level action decision and the low-level action function. This process involves multi-dimensional reasoning capabilities, which is referred to as hybrid-capabilities reasoning. Ideally, we want to activate the experts with the corresponding capability conditioned on the current reasoning stage to generate the output token, hence, this represents an output-oriented activation.

The reason why MoE can not achieve output-oriented activation can be attributed to the auto-regressive mechanism, which forces the same token in the input sequence routed to the same expert when generating tokens in different reasoning stages. As shown in Figure 7, although we can force

the input tokens in prompt and screen summary to be routed to the specialized expert $\mathcal{E}_{\text{sp}}$ to generate the output token in the subtask plan stage, when generating output tokens in action decision stage, these input tokens will be forwarded to the $\mathcal{E}_{\text{sp}}$ again instead of the $\mathcal{E}_{\text{ad}}$. Thus, we can not use $\mathcal{E}_{\text{ad}}$ to process these preceding tokens, preventing the fully leveraging of capability of $\mathcal{E}_{\text{ad}}$ in action decision.

## B  DPO DATA SELECTION

To construct DPO data pairs, we sample $K$ reasoning trajectories $\mathcal{T}$ for each action. These trajectories fall into three categories, and we adopt a tailored pairing strategy for each using the reward $\mathcal{R}_{\text{IG}}$, $\mathcal{R}_{\text{IG}}^+$, $\mathcal{R}_{\text{ACC}}$ and $\mathcal{R}_{\text{CoT}}$:

**The sampled trajectories are all correct.** We partition the sampled trajectories by whether $\mathcal{R}_{\text{IG+}} = 1$. From the $\mathcal{R}_{\text{IG+}} = 1$ subset, we select the trajectory with the highest $\mathcal{R}_{\text{CoT}}$ as the chosen output, which represents the most effective reasoning trajectory leading to the correct result. Conversely, from the $\mathcal{R}_{\text{IG+}} = 0$ subset, we pick the trajectory with the lowest $\mathcal{R}_{\text{IG}}$ as the rejected output, to suppress the trajectory that reaches a correct answer while through invalid intermediate reasoning steps.

$$\text{chosen: } \arg\max\big\{\mathcal{R}_{\text{CoT}}(T^{(k)}) \,|\, T^{(k)} \in \mathcal{T}, \mathcal{R}_{\text{IG}}^+(T^{(k)}) = 1\big\}$$
$$\text{rejected: } \arg\min\big\{\mathcal{R}_{\text{IG}}(T^{(k)}) \,|\, T^{(k)} \in \mathcal{T}, \mathcal{R}_{\text{IG}}^+(T^{(k)}) = 0\big\} \tag{12}$$

**The sampled trajectories are partially correct.** We partition the sampled trajectories by whether $\mathcal{R}_{\text{ACC}} > 0$. From the $\mathcal{R}_{\text{ACC}} > 0$ subset, we prioritize selecting the trajectory with the highest $\mathcal{R}_{\text{CoT}}$ and $\mathcal{R}_{\text{IG+}} = 1$ as the chosen output, which represents the most effective reasoning trajectory leading to the correct result. From the $\mathcal{R}\text{ACC} = 0$ subset, we select the trajectory with the lowest $\mathcal{R}\text{IG}$ as the rejected output, as it represents the worst reasoning and should be avoided.

$$\text{chosen: } \begin{cases} \arg\max\big\{\mathcal{R}_{\text{CoT}}(T^{(k)}) \,|\, T^{(k)} \in \mathcal{T}, \mathcal{R}_{\text{ACC}}^+(T^{(k)}) > 0, \mathcal{R}_{\text{IG}}^+(T^{(k)}) = 1\big\} \\ \quad \text{if}\big\{T^{(k)} \,|\, \mathcal{R}_{\text{IG}}^+(T^{(k)}) = 1\big\} \neq \phi \\ \arg\max\big\{\mathcal{R}_{\text{CoT}}(T^{(k)}) \,|\, T^{(k)} \in \mathcal{T}, \mathcal{R}_{\text{ACC}}^+(T^{(k)}) > 0, \mathcal{R}_{\text{IG}}^+(T^{(k)}) = 0\big\} \\ \quad \text{if}\big\{T^{(k)} \,|\, \mathcal{R}_{\text{IG}}^+(T^{(k)}) = 1\big\} = \phi \end{cases}$$
$$\text{rejected: } \arg\min\big\{\mathcal{R}_{\text{IG}}(T^{(k)}) \,|\, T^{(k)} \in \mathcal{T}, \mathcal{R}_{\text{IG}}^+(T^{(k)}) = 0\big\} \tag{13}$$

**The sampled trajectories are all wrong.** We use the ground truth reasoning trajectory $T^*$ as chosen output and choose the trajectory with the lowest $\mathcal{R}\text{IG}$ as the rejected output.

$$\text{chosen: } T^*$$
$$\text{rejected: } \arg\min\big\{\mathcal{R}_{\text{IG}}(T^{(k)}) \,|\, T^{(k)} \in \mathcal{T}\big\} \tag{14}$$

## C  PROGRESSIVE TRAINING STRATEGY

In order to empower CoME with hybrid-capabilities reasoning, we propose a progressive training strategy consists of three stages: (1) **Expert Finetuning (Expert-FT)**, which explicitly decouples and enhances different capabilities; (2) **Router Finetuning (Router-FT)**, which allows the activation of expert aligned with the current reasoning stage; (3) **Chain-of-Thought Finetuning (CoT-FT)**, which facilitates seamless collaboration and balanced optimization among experts. The overall training framework is shown in Figure 8.

## D  EXPERIMENT SETUP DETAILS

### D.1  DATASET

We train and evaluate CoME on both AITZ (Zhang et al., 2024) and AMEX (Zhang et al., 2024).

**AITZ** is a cleaned subset of the large scale AITW (Rawles et al., 2023), comprising 2.5K unique instructions and 18K steps. These instructions, drawn from over 70 mobile apps, are divided into

Figure 8: Overall training framework of CoME.

five categories: general, google apps, install, web shopping, and single. Each step is annotated with a chain-of-action-thought, covering screen summarization, action planning, action description, and action result.

**AMEX** is a large-scale mobile dataset comprising 104k screenshots and 3k instructions collected from 64 distinct apps. It features approximately 1.6M GUI element grounding annotations and 712k GUI element function descriptions. We further augment AMEX using the data construction methodology established by AITZ, the augmented data will be open-sourced.

## D.2 BASELINES

We compare CoME with two types of baselines: (1) Dense mobile agents and (2) Sparse MoE models.

**Dense mobile agents** are the general mobile agents pre-trained on large mobile datasets with strong capability on mobile environments, including:

• **Auto-GUI** (Zhang & Zhang, 2024) proposes a chain-of-action technique, that incorporates the text described previous action history and future action plan to facilitate action decision on the current screen.

• **SeeClick** (Cheng et al., 2024) only relies on screenshots for mobile task automation. SeeClick is pre-trained on large scale of mixed data including widget caption, UI summarization and UI grounding, together with general multi-modal dataset, *e.g.,* VQA and visual reasoning.

• **ShowUI** (Lin et al., 2024) uses UI-guide visual token selection to formulate screenshot as an UI connected graph, thus reducing the computational cost. The introduced interleaved vision-language-action streaming can flexibly manage visual-action history in navigation to enhance training efficiency.

• **UITars** (Qin et al., 2025) is a native GUI agent that solely perceives the screenshot as input and perform end-to-end action decision, which incorporates the unified action modeling and system-2 reasoning. UITars is pre-trained on large-scale screenshots with precise caption and grounding annotation. Moreover, it is iterative trained with reflective online traces to continuously learn from its mistakes.

• **UGround** (Gou et al., 2024) is trained with a large scale of 10M GUI elements and the target bounding box over 1.3M screenshots to empower strong grounding ability on mobile environment.

• **OS-Atlas** (Wu et al., 2024b) is a foundational GUI action model with grounding mode, action mode and agent mode. It can generate an action description as a simple plan and grounding the action on the screenshot, which excels at GUI grounding and OOD agentic tasks.

• **SphAgent** (Chai et al., 2024) is pre-trained on the AMEX dataset, including three-level annotations: interactive element grounding, screen and element functionality descriptions, and instruction with action chains. SphAgent is equipped with strong capabilities on screen understanding and element recognition.

**Sparse MoE models** are the MoE base models that are pretrained on the general visual-language tasks and fine-tuned on the mobile environment by us, including:

• **MolmoE** (Deitke et al., 2024) is a multi-modal MoE MLLM with 1.5B active and 7.2B total parameters, which nearly matches the performance of GPT-4V on both academic benchmarks and human evaluation.

• **DeepSeekVL2** (Wu et al., 2024c) incorporates a dynamic tiling vision encoding strategy to process high-resolution images and leverages DeepSeekMoE (Dai et al., 2024) with Multi-head Latent Attention and shared experts, which has only 4.5B activated parameters.

• **AriaUI** (Yang et al., 2024) is the first native MoE-based mobile agent that leverage textual or text-image interleaved action from trajectory to enhance the dynamic contexts understanding.

### D.3 METRICS

Following Auto-GUI (Zhang & Zhang, 2024), we use **action type** score to evaluate the accuracy of the predicted action type as well as the **action match** score to measure the accuracy of the predicted action parameter. For CLICK action, the distance between predicted action and labeled action less than 140 on the normalized 0–1000 coordinate scale is considered correct, or both of the clicked coordinate fall in the same bounding box. For TYPE action, the F1 score between predicted text and labeled text greater than 0.5 is considered correct. For other actions, we use exact match between the predicted action and labeled action.

### D.4 IMPLEMENTATION DETAILS

CoME comprises four distinct experts, each initialized from the Feed-Forward Network (FFN) layers of Qwen2VL-2B. During the Expert-FT stage, we exclusively fine-tune the FFN layers while freezing all other parameters. Training is performed on task-specific subsets—screen summary, subtask plan, action decision, and action function—extracted from the original hybrid-capabilities reasoning data of AITZ and AMEX. This stage is conducted using a cosine learning rate scheduler with a peak learning rate of 1e-5. In the Router-FT stage, we annotate each output token according to its corresponding reasoning stage and expert. Only the channel router is trained in this stage, using a peak learning rate of 1e-4 with a cosine scheduler. During the CoT-FT stage, we fine-tune the language model component of CoME while freezing the Vision Transformer (ViT). We adopt a LoRA configuration with rank 64 and LoRA alpha 128. This stage uses a peak learning rate of 2e-4 and follows a cosine learning rate schedule. For Info-DPO, we sample $K = 10$ reasoning trajectories for each step. And regarding the thresholds in Eq. 10, we set the distance threshold $\delta_d$ to 50 and the function threshold $\delta_f$ to 0.5. During Info-DPO training, we maintain the same configuration as CoT-FT but reduce the peak learning rate to 1e-4. All training is conducted on 8*A100 (80GB) with the total batch size of 128 for 5 epochs. For the weights of auxiliary SFT loss and Router Norm loss in Eq. 6 and Eq. 11, we set $\alpha$ to 1 and $\beta$ to 0.1.

## E ANALYSIS OF RELATIVE IMPROVEMENT RESULTS ON AITZ

We conducted a more detailed analysis of the performance improvements across all methods. Specifically, we computed the average performance of all methods and treated it as a new baseline (Average). We then measured each method's improvement relative to this baseline. Additionally, we calculated the variance of accuracy improvements across different action categories. The results are shown in the Table 8. Only four methods achieved consistent performance improvements across all action types. Among them, CoME demonstrated the highest relative improvement (+11.56%) while maintaining a lower improvement bias (4.41). Furthermore, we observed that methods incorporating multiple experts, such as MoE and CoME, exhibit lower improvement bias across different action categories compared to Dense models. This indicates that leveraging specialized experts not only enhances overall performance but also contributes to greater robustness of the method.

Table 8: Detailed results on AITZ with relative improvement and improvement variance.

| Method | SCROLL | CLICK | TYPE | PRESS | STOP | OVERALL | IMPROVE |
|---|---|---|---|---|---|---|---|
| Average | 43.57 | 51.92 | 74.15 | 57.74 | 69.07 | 55.41 | - |
| AutoUI | $61.40_{+17.83}$ | $32.20_{-19.72}$ | $81.40_{+7.25}$ | $57.70_{-0.04}$ | $74.40_{+5.33}$ | 47.69 | $-7.73_{12.37}$ |
| Qwen2VL(2B) | $23.05_{-20.52}$ | $43.74_{-8.18}$ | $59.40_{-14.75}$ | $52.22_{-5.52}$ | $47.02_{-22.05}$ | 43.80 | $-11.62_{6.53}$ |
| ShowUI | $23.79_{-19.78}$ | $40.34_{-11.58}$ | $85.20_{+11.05}$ | $59.00_{+1.26}$ | $87.55_{+18.48}$ | 49.55 | $-5.87_{14.07}$ |
| UITars(2B) | $30.17_{-13.40}$ | $55.25_{+3.33}$ | $84.77_{+10.62}$ | $53.78_{-3.96}$ | $61.39_{-7.68}$ | 55.63 | $+0.21_{8.40}$ |
| SeeClick | $11.14_{-32.43}$ | $52.96_{+1.04}$ | $53.00_{-21.15}$ | $67.88_{+10.14}$ | $55.36_{-13.71}$ | 49.11 | $-6.31_{15.23}$ |
| Qwen2VL(7B) | $43.28_{-0.29}$ | $55.30_{+3.38}$ | $51.40_{-22.75}$ | $56.92_{-0.82}$ | $64.87_{-4.20}$ | 54.46 | $-0.96_{9.22}$ |
| UIGround | $58.22_{+14.65}$ | $58.48_{+6.56}$ | $73.85_{-0.30}$ | $58.22_{+0.48}$ | $68.78_{-0.29}$ | 60.19 | $+4.77_{5.81}$ |
| OS-Atlas | $76.12_{+32.55}$ | $54.83_{+2.91}$ | $88.60_{+14.45}$ | $68.67_{+10.93}$ | $81.75_{+12.68}$ | 65.11 | $+9.69_{9.75}$ |
| UITars(7B) | $56.50_{+12.93}$ | $63.87_{+11.95}$ | $85.77_{+11.62}$ | $58.22_{+0.48}$ | $71.29_{+2.22}$ | 65.41 | $+9.99_{5.34}$ |
| MolmoE | $28.19_{-15.38}$ | $33.17_{-18.75}$ | $62.00_{-12.15}$ | $37.07_{-20.67}$ | $54.96_{-14.11}$ | 38.23 | $-17.19_{3.95}$ |
| Qwen2VL-MoE | $48.75_{+5.18}$ | $59.43_{+7.51}$ | $74.00_{-0.15}$ | $57.70_{-0.04}$ | $70.83_{+1.76}$ | 60.69 | $+5.27_{3.02}$ |
| AriaUI | $53.73_{+10.16}$ | $60.20_{+8.28}$ | $80.80_{+6.65}$ | $63.70_{+5.96}$ | $76.38_{+7.31}$ | 63.56 | $+8.14_{1.46}$ |
| CoME | $52.07_{+8.50}$ | $65.22_{+13.30}$ | $83.80_{+9.65}$ | $59.53_{+1.79}$ | $83.33_{+14.26}$ | 66.98 | $+11.56_{4.41}$ |

## F ANALYSIS OF DPO DATA SELECTION STRATEGIES.

For each action, we sample ten reasoning trajectories and construct DPO pairs using one of three strategies as described in Appendix B: (1) **cc**: pair two correct trajectories, if the sampled trajectories are all correct; (2) **cw**: pair a correct trajectory with a wrong one, if the sampled trajectories are partially correct; and (3) **lw**: pair the label with an wrong trajectory, if the sampled trajectories are all wrong. We provide more detailed analysis of information gain comparison among different combination of the pairing strategy as shown in Table 9. Higher InfoGain and InfoPositive will lead to higher action accuracy, because the higher InfoGain indicates that the reasoning trajectory provides more useful information, and higher InfoPositive shows that the intermediate reasoning step is much more valid. Introducing pair data from lw will decrease the InfoGain in reject output, because lw shows that the task is difficult for CoME to finish, the trajectory with the lowest $\mathcal{R}_{IG}$ contains less useful information. Introducing pair data from cc increase the InfoGain in reject output, because some suboptimal correct trajectories are added to the rejected output.

Table 9: Analysis of data selection strategy

| Strategy | IG@SS | | IG@AP | | IG@AD | | InfoGain | | InfoPositive | | Action Accuracy | |
|---|---|---|---|---|---|---|---|---|---|---|---|---|
| | reject | choose | reject | choose | reject | choose | reject | choose | reject | choose | type | match |
| AITZ | | | | | | | | | | | | |
| cc+cw+lw | 0.0748 | 0.0234 | -0.1430 | 0.1595 | -0.4354 | 0.0814 | -0.5037 | 0.2645 | 0.0291 | 0.6115 | 77.60 | 65.96 |
| cc+cw | 0.0762 | **0.0908** | -0.1294 | 0.1125 | -0.4252 | 0.0689 | -0.4784 | **0.2723** | 0.0300 | **0.7045** | 78.60 | 66.98 |
| cw+lw | 0.0686 | -0.0103 | -0.2049 | **0.1790** | -0.5758 | **0.0905** | -0.7121 | 0.2592 | 0.0380 | 0.4631 | 76.88 | 64.07 |
| cw | 0.0695 | 0.0808 | -0.1973 | 0.1143 | -0.5869 | 0.0741 | -0.7147 | 0.2693 | 0.0409 | 0.5654 | 77.75 | 65.93 |
| AMEX | | | | | | | | | | | | |
| cc+cw+lw | 0.0178 | 0.0368 | -0.1735 | **0.1373** | -0.3803 | 0.0327 | -0.5360 | 0.2068 | 0.0159 | 0.4824 | 83.79 | 70.04 |
| cc+cw | 0.0175 | **0.0385** | -0.1629 | 0.1359 | -0.3630 | 0.0337 | -0.5085 | **0.2083** | 0.0172 | **0.5111** | 84.46 | 72.61 |
| cw+lw | 0.0054 | 0.0172 | -0.2268 | 0.1293 | -0.4257 | 0.0342 | -0.6470 | 0.1808 | 0.0183 | 0.3797 | 84.44 | 71.14 |
| cw | 0.0038 | 0.0166 | -0.2199 | 0.1266 | -0.4103 | **0.0357** | -0.6265 | 0.1790 | 0.0199 | 0.4004 | 84.92 | 72.15 |

## G ANALYSIS OF REWARD STRATEGIES

We provide detailed analysis of different strategies in Info-DPO, including different reward model scales and reward function design in Table 10. As for the reward model scale, using 7B reward model can achieve comparable and even better overall performance compared with 72B reward model. While 72B reward model performs much better on CLICK action, because click action is much more challenging that requires precise description of the element to click and coordinates of the element

grounding. However, 72B reward results in lower InfoGain which is highly relevant to the action accuracy, because 72B reward model have a better fitness and lower cross-entropy loss, thus the difference of information entropy between different stage is much more smaller. As for different reward function designs, the continuous reward function proves to be more effective, particularly on the CLICK action. This is because a smaller distance between the predicted and ground-truth coordinates indicates higher prediction accuracy, and such outputs should be more likely selected as the chosen output for model optimization. However, this level of granularity cannot be captured by discrete reward functions.

Table 10: Analysis of reward strategy

| Method | ACTION ACCURACY | | | | | | REWARD MARGIN | | | |
|---|---|---|---|---|---|---|---|---|---|---|
| | SCROLL | CLICK | TYPE | PRESS | STOP | OVERALL | IG@SS | IG@AP | IG@AD | InfoGain |
| Reward Model Scale | | | | | | | | | | |
| Reward-7B | 52.07 | 65.22 | 83.80 | 59.53 | 83.33 | 66.98 | 0.0113 | 0.3117 | 0.6610 | 0.9841 |
| Reward-72B | 53.06 | 67.37 | 82.20 | 54.57 | 74.60 | 66.85 | 0.0087 | 0.1681 | 0.6597 | 0.8365 |
| Reward Function Design | | | | | | | | | | |
| Continuous | 52.07 | 65.22 | 83.80 | 59.53 | 83.33 | 66.98 | 0.0113 | 0.3117 | 0.6610 | 0.9841 |
| Discrete | 54.71 | 62.73 | 83.40 | 61.36 | 78.05 | 65.51 | 0.0143 | 0.2379 | 0.5018 | 0.7541 |

## H ANALYSIS OF TRAINING REWARD MODEL

InfoGain reward is estimated based on the difference in information entropy before and after introducing a specific reasoning stage. This estimation leverages a capability that language models naturally possess. Ideally, any logically consistent reasoning should make it easier for a model to predict the correct action. Therefore, we compare the effectiveness of training-based and training-free reward models, as shown in the Table 11. Training-free reward model could achieve comparable performance with training-based one, indicating that the estimation of InfoGain reward is a inner capability of the LLM with the strong general language modeling ability. Experimental results also demonstrate that InfoGain reward is a robust method to evaluate the CoT reasoning process.

Table 11: Comparison of training reward model

| Reward Model | Reward Acc | Action Acc |
|---|---|---|
| training-free | 82.24 | 66.64 |
| training-based | 84.68 | 66.98 |

## I ANALYSIS OF EFFICIENCY

### I.1 ANALYSIS OF GPU MEMORY CONSUMPTION

In this section, we present a more comprehensive analysis of GPU memory consumption during inference. Specifically, we compare the inference computational cost and action accuracy across three representative architectures: a 7B dense model, a 2B×8 MoE model, and our proposed CoME. The results, summarized in Table 12, demonstrate that CoME achieves the best overall trade-off, delivering the highest action accuracy while maintaining the lowest GPU memory footprint. The key memory advantage of CoME lies in its architecture design: the parameter set dominates the overall memory usage, whereas the additional KV cache overhead incurred during inference remains relatively minor. Taken together, these results underscore that CoME not only achieves best accuracy but also remains resource-efficient.

### I.2 ANALYSIS OF COMPUTATIONAL COST

Table 12: Detailed analysis of GPU memory usage

|  | Model | Extra | Total | Acc |
|---|---|---|---|---|
| Qwen2VL-7B | 15.48 GB | 1.04 GB | 16.52 GB | 54.46 |
| Qwem2VL-MoE | 21.34 GB | 0.98 GB | 22.32 GB | 60.69 |
| UITars | 15.49 GB | 1.43 GB | 16.92 GB | 65.41 |
| CoME | 10.55 GB | 1.14 GB | 11.69 GB | 66.98 |

In this section, we provide a theoretical analysis of the computational cost. We compare the FLOPs of CoME with strong baseline UITars at each layer. We denote $H$ as the hidden size, $L$ as the sequence length, and $I$ as the FFN's intermediate size of CoME. From the model's configuration files, we can find that:

$$H_{\text{UITars}} \approx 2.33H, \quad I_{\text{UITars}} \approx 2.11I$$

In the standard transformer layer, the computation cost of attention layer and FFN layer can be estimated by:

$$\text{Cost}_{\text{Attn}} = 4LH^2 + 2L^2H$$
$$\text{Cost}_{\text{FFN}} = 2LHI$$

Thus, for CoME, we forward the hidden states from different channels in parallel to the attention layer and the expert layer:

$$\text{Cost}_{\text{CoME}} = 4 \times (4LH^2 + 2L^2H) + 4 \times 2LHI$$
$$= 16LH^2 + 8L^2H + 8LHI$$

For UITars:

$$\text{Cost}_{\text{UITars}} = 4 \times (4L(2.33H)^2 + 2L^2(2.33H)) + 4 \times 2L(2.33H)(2.11I)$$
$$= 21.71LH^2 + 4.66L^2H + 9.83LHI$$

This result shows that CoME has lower computational cost than UITars at the FFN layer. While at the attention layer, as the sequence length in our settings is smaller than the hidden size, the first item is dominant and CoME also has lower computational cost at attention layer.

# J CASE STUDIES

## J.1 CASE STUDY OF DPO DATA PAIR

In Figure 9, we present the InfoGain values at each stage where the correct output is chosen and the wrong output is rejected. It can be observed that the chosen output exhibit InfoGain greater than zero at each stage, indicating that the reasoning in each stage contributes positively to predicting the correct action. For the rejected outputs, however, during the subtask planning stage the model produced an incorrect next action plan, resulting in negative InfoGain. This shows that the reasoning at this stage has a detrimental impact on predicting the correct action. This case illustrates that InfoGain can be used to evaluate the contribution of intermediate reasoning steps to the prediction of the correct action.

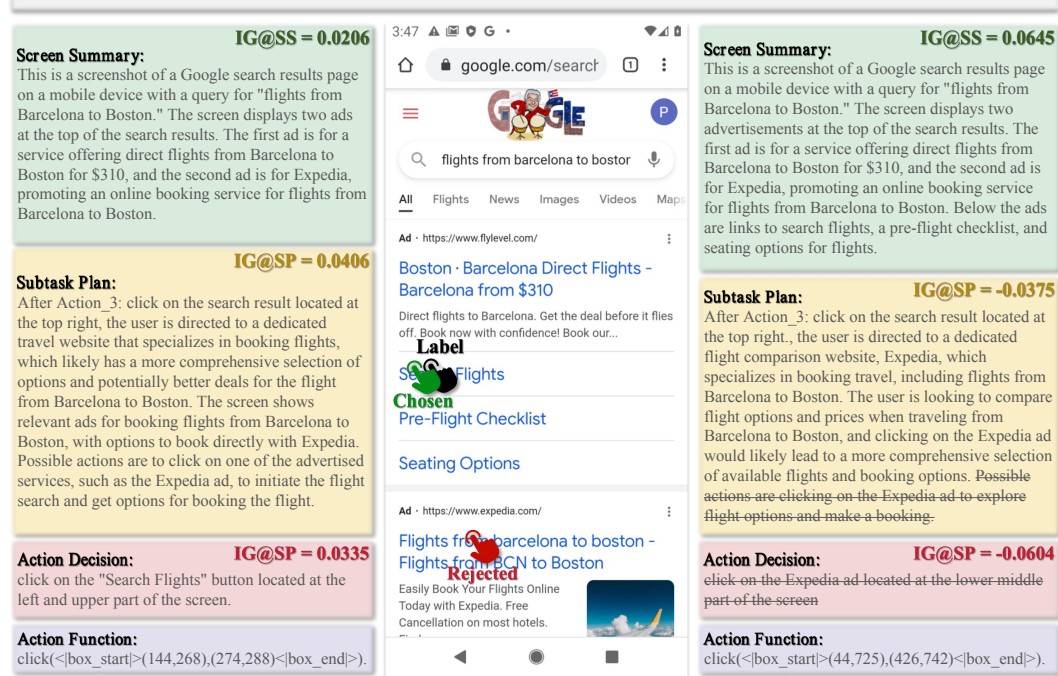

Figure 9: InfoGain of DPO data pair where the correct one is chosen and the wrong one is rejected.

In Figure 10, we present the InfoGain values at each stage where both the chosen and rejected outputs are correct. Although the rejected output also result in the correct action, it encounters some reasoning mistakes in subtask plan and action decision stage. Thus the InfoGain at these two stages are negative, indicating that the reasoning process in that stage may be lead to the wrong action. This case illustrates that InfoGain can effectively identify the incorrect intermediate reasoning steps thus suppressing these ineffective reasoning pattern. While this is impossible for the outcome-level reward used by naive DPO.

You are an intelligent agent that can help to automated finish the given task. You are given the current screenshot and a task to finish. You should step-by-step analyze how to finish the task.

The given task is: **uninstall "PUBG MOBILE"**

Your previous actions are:
Action_0: press the home button
Action_1: scroll up
Action_2: scroll up
Action_3: click on the Play Store located at lower left.
Action_4: click on the \"Search\" icon located at the top-right corner.

**IG@SS = 0.2535**

**Screen Summary:**
This screenshot shows a smartphone display of a search results page, likely within an app store or search engine, with queries related to "Google Docs". The top result is "Google Docs" itself, followed by related search terms such as "document editor" and "google drive", each accompanied by a magnifying glass icon and a rightward-pointing arrow. The user interface suggests the user can tap on any of these search suggestions to perform a specific search

**IG@SP = 0.2898**

**Subtask Plan:**
After Action_4: click on the "Search" icon located at the top-right corner., the search function has been activated, allowing for the input of the term "PUBG MOBILE" to locate the application within the device's application settings or store. The current search term "Google Docs" is not relevant to the task of uninstalling "PUBG MOBILE"; clearing the search field is a step to correct the search query. Possible actions are clicking on the "X" to clear the current search results.

**IG@SP = 0.0328**

**Action Decision:**
click on the clear icon "X" located at the top right.

**Action Function:**
click(<|box_start|>(912,46),(943,73)<|box_end|>).

**IG@SS = 0.2323**

**Screen Summary:**
This screenshot shows a smartphone display of a search results page, likely within an app store or a search feature within an app, with a focus on Google Docs and related services. The top search result is "Google Docs" with additional related searches listed below, such as "document editor," and "google drive," and there are icons suggesting the device is connected to Wi-Fi and has a sufficient battery level. The interface allows the user to click on any of these related searches.

**IG@SP = -0.0965**

**Subtask Plan:**
After Action_4: click on the "Search" icon located at the top-right corner., the search functionality is activated, allowing for the input of the application name "PUBG MOBILE" to locate the app within the device's application store or settings. The current search term "Google Docs" is not relevant to the task of uninstalling "PUBG MOBILE". ~~Possible actions are to click on the search field to edit the text~~ from "Google Docs" to "PUBG MOBILE" in order to find the correct application and proceed with its uninstallation.

**IG@SP = -0.1788**

**Action Decision:**
~~click on the search icon located at the top right~~

**Action Function:**
click(<|box_start|>(900,46),(923,73)<|box_end|>).

Figure 10: InfoGain of DPO data pair where both the chosen and rejected outputs are correct.

## K   LLM USAGE STATEMENT

This paper employed an LLM solely to refine our manually written draft, including improving word choice, grammar correctness, and sentence fluency.

