# OpenReview forum: "Empowering Channel-of-Mobile-Experts with Informative Hybrid-Capabilities Reasoning"
_ICLR.cc/2026/Conference — Submitted to ICLR 2026_

### Official Review · Reviewer_rUrh · 2025-10-17

**Soundness:** 3
**Presentation:** 3
**Contribution:** 3
**Rating:** 6
**Confidence:** 2

**Summary:**

The paper introduces Channel-of-Mobile-Experts, an agent architecture that assigns different experts to distinct reasoning stages and uses output-oriented activation instead of conventional MoE routing. A progressive training pipeline and an InfoGain-driven DPO strategy are proposed to enhance capability specialization and reduce error propagation. Experiments on AITZ and AMEX show that CoME outperforms dense and sparse baselines in both action type and action match accuracy, with supporting ablation studies validating the design choices.

**Strengths:**

1. The paper introduces a well-motivated architectural design that addresses limitations of conventional MoE routing in multi-stage reasoning.The progressive training strategy is clearly structured and each stage is supported by ablation results. The InfoGain-driven DPO provides a principled way to enhance intermediate reasoning quality and reduce error propagation.
2. The experiments are extensive, covering two representative mobile benchmarks with both action type and action match evaluation, and the results demonstrate consistent improvements over dense and sparse baselines.
3. The visualizations and analysis (e.g., expert activation patterns, InfoGain distributions, ablation breakdowns) are clear and intuitive, which helps justify the design choices.

**Weaknesses:**

1. The paper targets mobile agents, but the efficiency aspect is not sufficiently addressed. In particular, there is no comparison in terms of FLOPs or computational cost, which is important for evaluating mobile settings and ensuring fairness across baselines.

2. here are a few minor typos in the paper. For example, in Section 3.4, Eq. (6), “Nrom” should be “Norm”.

**Questions:**

See weakness

---

> ### Author Response · Authors · 2025-11-21
> **Response to Reviewer rUrh**
>
> We sincerely thank you for your insightful suggestions and valuable comments! We will try to address your concerns in detail, and we would appreciate it very much if you could kindly raise your score if your concerns are addressed.
>
> ---
>
> > W1: comparison in terms of FLOPs
>
> Thank you for raising this question. We also agree that for mobile scenarios, the computational cost of the model is very important. Therefore, we compared our approach with the strong baseline UITars.
>
> We denote **H** as the hidden size, **L** as the sequence length, and **I** as the FFN's intermediate size of CoME. From the model's configuration files, we can find that:
>
> $$
> H_{\text{UITars}} \approx 2.33 H, \quad I_{\text{UITars}} \approx 2.11 I
> $$
>
> And we compare the FLOPs of UITars and CoME at each layer:
>
> |        | Action Match | FLOPs               |
> |--------|--------------|---------------------|
> | UITars | 65.41        | $21.71LH^2+4.66L^2H+9.83HLI$ |
> | CoME   | 66.98        | $16LH^2+8L^2H+8LHI$  |
>
> This result shows that CoME has lower computational cost than UITars at the FFN layer. While at the Attention layer, as the sequence length in our settings is smaller than the hidden size, the first item is dominant and CoME also has lower computational cost at Attention layer. Despite this, CoME achieves better performance.
>
>
> > W2: typos in the papers
>
> Thank you for your suggestion. We will carefully review the manuscript and correct any existing typos in the revision.

---

> > ### Comment · Reviewer_rUrh · 2025-11-22
> >
> > Thank you for the detailed and thoughtful responses. I appreciate the clarifications and the additional analysis provided by the authors.

---

> > > ### Author Response · Authors · 2025-11-22
> > > **Response to Reviewer rUrh**
> > >
> > > Thank you for taking the time to read our responses and for your timely reply. Your questions and suggestions are very valuable; accordingly, we have provided additional analysis and revised our paper. We would appreciate knowing whether they have resolved your concerns. If so and you are satisfied, we would be grateful if you would consider raising your score. Thanks once again for your effort in reviewing our work!

---

### Official Review · Reviewer_7j82 · 2025-10-29

**Soundness:** 2
**Presentation:** 3
**Contribution:** 2
**Rating:** 4
**Confidence:** 4

**Summary:**

This paper tackles the challenge of hybrid-capabilities reasoning in mobile agents - the multi-stage process of understanding screens, planning subtasks, deciding actions, and executing functions when following user instructions. Current approaches either focus on individual capabilities in isolation or use large-scale pre-training that leads to imbalanced performance. Traditional MoE models, while supporting capability decoupling, use input-oriented activation that fails to align experts with specific reasoning stages.
The authors propose Channel-of-Mobile-Experts (CoME), featuring output-oriented activation where experts are selected based on the current reasoning stage rather than input features. CoME employs four specialized experts with a channel router for dynamic selection, trained through a progressive three-stage strategy (Expert-FT, Router-FT, CoT-FT). Additionally, InfoGain-Driven DPO quantifies each intermediate step's contribution to mitigate error propagation.

**Strengths:**

- The paper is well-motivated and easy to read.
- The paper provides detailed analysis from multiple perspectives including expert activation distribution, InfoGain rewards, and efficiency.

**Weaknesses:**

- The fixed four experts (screen summary, subtask planning, action decision, action function) may limit extensibility to other domains or tasks.
- The experimental performance does not demonstrate significantly superior results compared to baselines. For instance, Auto-GUI achieves comparable performance to the proposed methodology despite using only 700M parameters versus 5B parameters.
- In Section 4.4.3, comparisons should include not only Qwen but also other baselines that show comparable performance.

**Questions:**

- Is there a way to train the router in a self-supervised manner without expert labels for task?
- How does CoME handle tasks where reasoning stages are not clearly defined?

---

> ### Author Response · Authors · 2025-11-21
> **Response to Reviewer 7j82 (1/2)**
>
> We sincerely appriciate for your constructive feedback and insightful comments. We will address each of your questions in detail, and we would be most grateful if you might consider raising your score once your concerns have been fully resolved.
>
> ---
>
> > W1 & Q2: extensibility to other domains or tasks
>
> Thanks for your questions. First, we would like to clarify the motivation behind our CoME design. Our decomposition of the reasoning process is directly inspired by the mainstream Agent paradigm ReAct [1], which structures reasoning into observation, thought and action. This design has been shown to help agents interpret the current environment, perform deliberate intermediate reasoning, and produce more accurate actions—an approach that has proven broadly effective in general agent tasks.
> Importantly, we find that **ReAct paradigm is also highly suitable for mobile-agent scenarios**. In our setting, observation corresponds to describing the current screen state and contextual information, while thought captures how the model analyzes task progress and plans subsequent steps. A similar chain-of-thought structure is also adopted in prior works such as UITars [2] and OS-ATLAS [3], where they also include a natural-language description of the intended action before converting it into an executable action function, leading to better action prediction.
> Building on these widely validated findings, we design our four-stage reasoning framework following the ReAct format. Our experiments on Qwen2VL further confirm that **explicitly structuring the reasoning process into stages yields better performance**. This indicates that the explicit stage formulation provides the model with clearer guidance on how to conduct reasoning, leading to more reliable and effective behavior.
>
> |                  | Action Type | Action Match |
> |------------------|-------------|--------------|
> | raw reasoning    | 70.86       | 52.98        |
> | staged reasoning | 71.59       | 54.46        |
>
> In addition, ReAct paradigm shows that different reasoning stages emphasize different model capabilities. Building on this insight, we designed Channel-of-Mobile-Experts (CoME), which consists of four experts, each specializing in a distinct type of capability. Through our proposed output-oriented activation mechanism, CoME ensures that each expert is activated precisely at the reasoning stage where its capability is most relevant, achieving effective stage–expert alignment.
> Although some tasks do not exhibit explicit reasoning stages, **these tasks can still be naturally associated with specific capability**. In such cases, CoME only needs to activate the corresponding expert(s), allowing it to handle these tasks without requiring additional structural modifications.
> Following the your suggestion, we further evaluated CoME on tasks that don't require explicit reasoning stage—such as Captioning, OCR, and Grounding. The results demonstrate that **CoME achieves comparable or even better performance on these tasks**, confirming that our design generalizes across diverse reasoning requirements.
>
> |            | Caption F1 | OCR F1 | Grounding Acc |
> |------------|------------|--------|---------------|
> | Qwen2VL-7B | 54.53      | 78.87  | 78.89         |
> | UItars-7B  | 54.08      | 78.87  | 79.11         |
> | MoE-A3B    | 52.34      | 78.75  | 77.32         |
> | CoME-5B    | 53.89      | 79.67  | 79.19         |

---

> ### Author Response · Authors · 2025-11-21
> **Response to Reviewer 7j82 (2/2)**
>
> > W2: improvement compared to baselines
>
> Thank you for raising this concern. To address it, we conducted evaluations ten times on both datasets and assessed statistical significance using a t-test. On AITZ, CoME achieves 66.89 ± 0.14, which **significantly outperforms the strongest baseline by 1.57 under p < 0.01**. Similarly, on AMEX, CoME obtains 72.64 ± 0.16, again demonstrating a statistically significant improvement over the best baseline with p < 0.01.
> In addition, we would like to emphasize that **CoME achieves better performance while using both fewer training samples and fewer trainable parameters**. Below we provide a detailed comparison of the training data scale and parameter counts for completeness.
>
> | Method   | Params | Action Match | data statistic                    |
> |----------|--------|--------------|-----------------------------------|
> | UGround  | 7B     | 60.19        | 1.3M screens                      |
> | OS-Atlas | 7B     | 65.11        | 2.3M screens                      |
> | UI-Tars  | 7B     | 65.41        | million-level screens, 50B tokens |
> | CoME     | 5B     | 66.98        | 13k screens, 2k samples           |
>
> As for the performance of Auto-GUI, we would like to clarify that the action-type metric only reflects the model's ability to understand the task plan. It does not fully reflect the model's actual effectiveness. For example, once the user reaches a search-results page, it is indeed easy to infer that the next step should be a click operation. However, determining which search result to click is substantially more challenging. This discrepancy is also evident in the experiment results: although Auto-GUI achieves a 74% accuracy in click action type, its click action match drops sharply to 32%. In contrast, CoME achieves 83% in type accuracy and 65% in action match.
> Furthermore, a model's high-level task-planning ability can be learned relatively easily from patterns in large-scale training data. **Auto-GUI is trained on AITW, which is from the same data source as AITZ and contains millions of examples**, making its action-type accuracy is slightly higher. In comparison, CoME is trained only on ~10K AITZ samples, yet still surpasses Auto-GUI by +19.29 on the action-match metric.
>
> ---
>
> > W3: efficiency comparison with comparable baselines.
>
> Thank you for the suggestion. Since OS-Atlas and UITars are both built upon the Qwen2VL architecture, we initially did not include them in our comparison. However, considering that their data organization formats differ from ours, which may lead to slight differences in GPU memory consumption, we added comparisons against both OS-Atlas and UITars as recommended. **The results show that CoME achieves lower GPU memory overhead while also delivering better performance**, demonstrating its clear advantages over these methods.
>
> | method      | GPU MEM(train) | GPU MEM(infer) |
> |-------------|----------------|----------------|
> | Qwen2VL     | 31.62GB        | 16.52GB        |
> | OS-Atlas    | 30.49GB        | 16.44GB        |
> | UITars      | 32.08GB        | 16.92GB        |
> | Qwen2VL-MoE | 30.70GB        | 22.32GB        |
> | CoME        | 18.52GB        | 11.69GB        |
>
> ---
>
> > Q1: self-supervised router training
>
> Thank you for the insightful question. We experimented with removing expert labels and allowing the model to autonomously learn the alignment between expert activation and reasoning stages. However, our results show that this training strategy behaves similarly to MoE training: the model struggles to spontaneously achieve output-oriented activation, i.e., activating the expert whose capability corresponds to the specific reasoning stage needed to generate the output tokens.
> Therefore, at this stage, supervision with expert labels remains necessary for effective training. Your question raises an important direction, and we plan to further explore label-free expert activation mechanisms in future work.
>
> |                  | Action Type | Action Match | Expert Align |
> |------------------|-------------|--------------|--------------|
> | self-supervised  | 74.87       | 61.32        | 30.49        |
> | label-supervised | 75.10       | 62.93        | 98.81        |
>
> ---
>
> [1] ReAct: Synergizing Reasoning and Acting in Language Models
>
> [2] UI-TARS: Pioneering Automated GUI Interaction with Native Agents
>
> [3] OS-ATLAS: A Foundation Action Model for Generalist GUI Agents

---

> > ### Comment · Reviewer_7j82 · 2025-11-25
> >
> > Thank you for your response. I have an additional question regarding your response to Q1. If the model autonomously learns the expert activation stages, does this differ from the training approach of existing MoE methods? Could you please elaborate on what "autonomously learn the alignment" means in detail?

---

> ### Author Response · Authors · 2025-11-26
> **Response to Reviewer 7j82**
>
> Thank you for taking the time to read our responses and for your timely reply.
>
> ---
>
> We would like to provide a more detailed explanation of our response to Q1. In our "self-supervised" setting, we keep Stage-1 Expert-FT, remove Stage-2 Router-FT, and simplify Stage-3 CoT-SFT. In the simplified self-supervised CoT-SFT, we do not provide the expert label for each output token; instead, we allow the router to autonomously decide which expert to use for generating each token. For the training loss in Eq(6), we remove the $\mathcal{L}_{\text{R-Norm}}$ item. And in our Qwen2VL-MoE baseline, we also train the MoE model with the data from each expert domain first to let the MoE model implictly learn different expert abilities and also for fair comparison. As a result, the "self-supervised" training process is much  similar to the standard MoE approach.
>
> In fact, using expert labels to supervise the router does not incur substantial annotation cost. For each output token, we can automatically extract its expert label based on the reasoning stage it belongs to through a rule-based method. This makes the additional overhead almost negligible. Moreover, our results show that supervised training with expert labels is necessary: aligning expert activation with the reasoning stages improves reasoning performance, whereas the self-supervised approach fails to effectively learn such expert activation patterns.
>
> We acknowledge that your question highlights a vary valuable direction for the future research, which could provide a more flexible training paradigm for CoME and we alse plan to improve this training paradigm for expert activation in the future work. While, at the current stage, our focus is primarily on revealing the misalignment between expert activation and reasoning stages, and demonstrating that improving this alignment can indeed enhance reasoning performance, these findings could also illustrate the significance and contribution of our work.
>
> ---
>
> Thanks again for taking the time to review our work and for your valuable suggestions. We sincerely hope that our clarifications have addressed all of your concerns. If you find the responses satisfactory, we would truly appreciate it if you would consider raising your score. Should you have any remaining questions, we would be more than happy to engage in further discussion. Thank you once again.

---

### Official Review · Reviewer_bsc5 · 2025-10-29

**Soundness:** 3
**Presentation:** 2
**Contribution:** 3
**Rating:** 6
**Confidence:** 3

**Summary:**

This paper introduces Channel-of-Mobile-Experts (CoME), a transformer-based architecture for hybrid-capability mobile agents. Each layer duplicates hidden channels into E experts, each with its own FFN, and a channel router dynamically activates or blends experts depending on the current reasoning stage. Training proceeds through three progressive phases (Expert-FT 2 Router-FT 2 CoT-FT). To reduce error propagation in chain-of-thought reasoning, the authors propose InfoGain-Driven DPO, which uses a reward model to estimate the information gain (IG) of intermediate steps and selects trajectories accordingly. Experiments on AITZ and AMEX datasets show improvements over dense and MoE baselines, supported by ablation studies.

**Strengths:**

1. CoME’s design (channel repetition, per-expert FFN, router-based activation) and the three-phase finetuning are clearly described with formulas and pseudo-code, making replication feasible.
2. InfoGain-DPO uses reward-model-estimated information gain to filter and reweight DPO pairs, providing an intuitive way to reduce mid-stage reasoning errors.
3. The three-phase finetuning schedule (Expert-FT / Router-FT / CoT-FT) is conceptually neat and empirically validated through ablations.
4. The paper includes multiple baselines and ablation studies (e.g., w/o Info-DPO, w/o Router-FT, w/o Expert-FT), demonstrating consistent contributions across datasets, proving its strong performance.
5. Experiments show CoME yields consistent improvements on both image-centric and text-centric mobile tasks, suggesting robustness to input modality variation.

**Weaknesses:**

1. The effectiveness of InfoGain depends heavily on the reward model’s quality. The paper notes performance degradation when switching from a 7B to a 72B model but lacks a systematic robustness analysis.
Some key hyperparameters, such as DPO thresholds, sampling count K, and Router-Norm settings, are partially described or omitted from the appendix

**Questions:**

Please refer to the weakness for more details

---

> ### Author Response · Authors · 2025-11-21
> **Response to Reviewer bsc5**
>
> We sincerely appreciate your effort in reviewing our work and constructive suggestions. We will response to each of your questions in detail; if your concerns are resolved, we would be grateful for your continued support of our work and consider raising your score.
>
> ---
>
> > W1: Analysis of reward model quality
>
> Thank you for the constructive suggestions. Here we would like to clarify that InfoGain does not heavily depend on the reward model’s quality. **As shown in Table 11 in Appendix H**, we compare training-based and training-free reward models. The results indicate that the training-free reward model can achieve comparable performance, although the training-based version performs better. This demonstrates that InfoGain primarily relies on the model’s intrinsic logical understanding, and only requires the reward model to have this basic capability, rather than being heavily optimized.
>
> | Reward Model    | Reward Acc | Action Acc |
> |-----------------|------------|------------|
> | training-free   | 82.24      | 66.64      |
> | training-based  | 84.68      | 66.98      |
>
> Regarding the question about the weaker performance of the 72B reward model, **we have provided a detailed analysis in Appendix G**. By evaluating the InfoGain of each reasoning stage, we infer that the degradation stems from the overly strong fitting ability of the trained 72B reward model. After training, **72B reward model exhibits much lower cross-entropy loss, which makes it insufficiently sensitive to subtle errors in intermediate reasoning steps**. As a result, the distinction between positive and negative intermediate trajectories in the DPO data pair becomes less pronounced, leading to weaker DPO optimization signals and ultimately suboptimal performance.
>
> ---
>
> > W1: key hyperparameters
>
> Thank you for the helpful suggestion. We will add a more detailed description of the hyperparameter settings in the revision.
>
> Specifically, for InfoDPO, we use the following configurations:
> Information-gain thresholds: $\delta_d=50$, $\delta_f=0.5$;
> Number of sampled trajectories: $K=10$;
> For both CoT-SFT and Info-DPO, we set the Router-Norm weight to 0.01, which further constrains the alignment between expert activation and the corresponding reasoning stage.
>
> These details will be included in the revised Appendix to ensure full transparency and reproducibility.

---

> > ### Comment · Reviewer_bsc5 · 2025-11-24
> >
> > Thanks to the author's answers to my questions, almost all my doubts were resolved, so I will keep my original positive score.

---

> > > ### Author Response · Authors · 2025-11-26
> > > **Response to Reviewer bsc5**
> > >
> > > We sincerely appreciate your thorough review of our work and the many insightful suggestions you provided. We are pleased that our responses have successfully addressed your concerns. We would appriciate it if you could continue to support our work in the following stages. Thank your once again for your kindly engagement and timely response.

---

### Official Review · Reviewer_RhSw · 2025-10-30

**Soundness:** 3
**Presentation:** 4
**Contribution:** 2
**Rating:** 6
**Confidence:** 4

**Summary:**

This paper identifies two primary challenges for mobile agents:
* Hybrid-Capabilities Reasoning: Agents must sequentially perform multiple distinct tasks (screen summary, subtask planning, action decision, and action function generation). Existing models, like dense models or standard Mixture-of-Experts (MoE), struggle to decouple and balance these different capabilities effectively. The paper argues standard MoE fails because its "input-oriented activation" is unsuited for this stage-based reasoning.
* Error Propagation: In a multi-step reasoning chain, small errors in early stages (like a poor screen summary) can cascade and lead to an incorrect final action.

To solve this, the paper proposes two main contributions:
* Channel-of-Mobile-Experts (CoME): A novel agent architecture with four specialized experts, one for each of the four reasoning stages. It uses a "Channel Router" that employs "output-oriented activation" to intelligently select the correct expert's hidden states based on the current reasoning stage (e.g., it uses the 'subtask plan' expert when generating the subtask plan).
* InfoGain-Driven DPO (Info-DPO): A new training technique to improve credit attribution. Instead of just rewarding the final correct action, this method uses reward models to measure the "information gain" of each intermediate step. It then uses DPO to fine-tune the agent, teaching it to prefer reasoning trajectories where each step positively contributes to the final answer.

The CoME model is trained using a progressive strategy (Expert-FT, Router-FT, CoT-FT) and then refined with Info-DPO. The authors demonstrate that this approach outperforms existing dense and MoE-based mobile agents on the AITZ and AMEX benchmarks.

Main Results in the Paper
* State-of-the-Art Performance: CoME achieves the highest overall action match accuracy on both benchmarks (Tables 1 & 2).
* Component Importance (Ablation Study): The ablation study (Table 3) shows that removing Info-DPO causes the largest performance drop (a 4.68% average decrease), while removing the Router-FT stage (which trains the "output-oriented activation") also causes a severe performance drop.
* Architectural Validation: The paper shows why CoME works. Analysis (Figure 5) shows CoME's Channel Router has a ~99% accuracy in selecting the correct expert for the corresponding reasoning stage. In contrast, a standard MoE model activates experts almost randomly.
* Training Strategy Validation: The proposed Info-DPO is shown to be superior to a simpler "Rule-DPO" (Table 4). Info-DPO provides substantial gains, especially on the complex "CLICK" action, because it provides supervision on the intermediate reasoning steps, not just the final outcome.
* Efficiency: CoME is shown to be more memory-efficient than baselines (Table 7).

**Strengths:**

* Effective Solution for Credit Attribution / Error Propagation: The paper tackles the difficult, well-known problem of error propagation in Chain-of-Thought (CoT) reasoning. The InfoGain-DPO method is a novel and logical way to apply fine-grained rewards to intermediate reasoning steps, moving beyond final-action accuracy. The approach seems general and can be interesting / relevant for other domains.
* Strong Empirical Analysis: The paper presents very thorough validation of the results. The ablation studies are clear, and the visualization of the router's activation (Figure 5) provides some evidence that the proposed mechanism works as designed.

**Weaknesses:**

* Weaknesses in the CoME architecture: The proposed CoME architecture is not very convincing. First, it is not clear if it will be broadly applicable. The architecture is hard-coded with four experts, one for each stage of reasoning defined by the AITZ dataset. It is unclear how this rigid structure would adapt to different tasks or datasets that might have more, fewer, or different reasoning stages. It lacks the inherent flexibility of a more general MoE model. Second, looking at the empirical results, it is hard to see fair comparison. The comparison is against different architectures (MoE, dense) and different parameter counts. It is very hard to make sense of the results without fair iso-FLOP studies.

**Questions:**

* Are there other studies to understand the effectiveness of CoME architecture? One possible study could be to (pre-)train on the same dataset but with different architectures while matching the total FLOPs.
* Info-DPO seems interesting, are there any studies on other domains require long CoT reasoning (e.g., math or code)?

---

> ### Author Response · Authors · 2025-11-21
> **Response to Reviewer RhSw （1/2）**
>
> We sincerely thank you for your constructive suggestions and valuable comments! We will answer the questions in detail and would appreciate it very much if you could kindly consider raising your score if your concerns are addressed.
>
> ---
>
> > W1: CoME architecture is hard-coded with four experts and how to adapt to other tasks.
>
> Thanks for your questions. First, we would like to clarify that our design and segmentation of the reasoning stage and experts **follow the mainstream Agent paradigm, ReAct [1]**. ReAct decomposes the reasoning process into observation, thought, and action. This structure allows an agent to analyze the current environmental state, plan through deliberate thinking, and ultimately predict more accurate actions, which has proven effective for general agent tasks.
> Fortunately, we find that **ReAct paradigm is also well-suited to the mobile agent scenario**. In our setting, observation corresponds to describing the current screen state and contextual information, while thought corresponds to analyzing the task progress and determining subsequent plans. Moreover, the AITZ dataset happens to provide data that naturally matches this structure. *Similar CoT-style paradigms have also been adopted in prior works such as UITars [2] and OSAtlas [3]*, they also include a natural-language action description before converting it into an executable action function, thereby improving action prediction accuracy.
> Based on these widely validated findings, we designed our four-stage reasoning procedure following the ReAct format. Our experiments on Qwen2VL further confirm that **explicitly separating the reasoning process into distinct stages yields better performance** than using raw reasoning, as the model receives clearer guidance on how to structure its own reasoning process.
>
> |                  | Action Type | Action Match |
> |------------------|-------------|--------------|
> | raw reasoning    | 70.86       | 52.98        |
> | staged reasoning | 71.59       | 54.46        |
>
> In addition, we observed that different reasoning stages emphasize different aspects of the model’s capabilities. This motivated us to further conceptualize this reasoning paradigm as a form of hybrid-capability reasoning. To decouple these capabilities and optimize each of them more effectively, we proposed the Channel-of-Mobile-Experts (CoME) architecture, which includes four experts specializing in different reasoning abilities. Through output-oriented activation, CoME aligns expert activation with the corresponding reasoning stage. **CoME can also support versions with fewer or more experts**, we currently lack additional data covering more fine-grained reasoning stages, so we have not trained a "more-experts" variant. Considering your suggestion, we also explored the performance of CoME with fewer experts, where we removed the screen summary stage and used a 3-expert version of CoME. The performance on AITZ degrades, which further supports the necessity and rationality of our current four-stage reasoning design.
>
> |         | Action Type | Action Match |
> |---------|-------------|--------------|
> | 3-expert | 74.42       | 62.28        |
> | 4-expert | 75.10       | 62.93        |
>
> Following your suggestion, we also evaluated CoME on tasks that require additional types of reasoning stages, such as captioning, OCR, and grounding. The results show that CoME achieves comparable or even better performance on these tasks, further **demonstrating CoME's generality and robustness across different reasoning requirements**.
>
> |            | Caption F1 | OCR F1 | Grounding Acc |
> |------------|------------|--------|---------------|
> | Qwen2VL-7B | 54.53      | 78.87  | 78.89         |
> | UItars-7B  | 54.08      | 78.87  | 79.11         |
> | MoE-A3B    | 52.34      | 78.75  | 77.32         |
> | CoME-5B    | 53.89      | 79.67  | 79.19         |
>
> Finally, we would like to clarify that both CoME and MoE are flexible architectures. **The flexibility of CoME lies in its ability to dynamically adjust the architecture and the number of experts according to the capability requirements of the given task**. When a new capability is needed, a standard MoE architecture typically requires retraining all experts to adapt to this new ability. In contrast, CoME *only needs to train one new expert corresponding to the new capability and insert it into the original CoME structure*, followed by retraining the router. This makes CoME more friendly for scenarios that demand specialized capabilities, as the cost of adapting to new abilities is much lower.

---

> ### Author Response · Authors · 2025-11-21
> **Response to Reviewer RhSw （2/2）**
>
> > W1 & Q1: fair comparison that matches the total FLOPs
>
> Thank you very much for your suggestion. To provide a fair comparison, we implement an MoE model with 5B activated parameters, and the results show that **CoME outperforms the MoE with comparable amount of activated parameters**.
>
> |         | Action Type | Action Match |
> |---------|-------------|--------------|
> | MoE-A3B | 72.94       | 60.69        |
> | MoE-A5B | 75.06       | 60.94        |
> | CoME    | 75.10       | 62.93        |
>
> ---
>
> > Q2: Info-DPO on other domains
>
> Thanks for your insightful question. Although Info-DPO was originally designed for staged-reasoning scenarios in mobile agents, the core idea—using information gain to evaluate the effectiveness of intermediate reasoning—could in principle also apply to long CoT reasoning. We validated this on GSM8K, the experimental results show that **Info-DPO also achieves better performance than standard DPO on math domain**. This provides further evidence for the generality of Info-DPO beyond the mobile-agent setting.
>
> | method  | Acc   |
> |---------|-------|
> | SFT     | 64.37 |
> | DPO     | 66.03 |
> | InfoDPO | 66.59 |
>
> ----
>
> [1] ReAct: Synergizing Reasoning and Acting in Language Models
>
> [2] UI-TARS: Pioneering Automated GUI Interaction with Native Agents
>
> [3] OS-ATLAS: A Foundation Action Model for Generalist GUI Agents

---

### Meta-Review · Area_Chair_11Jz · 2026-01-04

**Summary:**

1. Hard-coded 4 experts tied to AITZ’s stage structure (screen summary → subtask plan → action decision → action function). Reviewers questioned whether this design would transfer to tasks with different / fewer / more / unclear stages. (RhSw, 7j82)

2. CoME seems less inherently flexible than standard MoE, where experts can emerge more organically from input-oriented routing. (RhSw, 7j82)

3. It was hard to interpret the gains without fair comparisons controlling for compute/parameters/FLOPs. and the improvements are marginal.

I think this paper is without enough technical contributions and rigorous evaluation framework to demonstrate the effectiveness of the proposed methods.

**Reviewer Concerns:**

I think the main remaining concern is the performance, the proposed method does not convincingly better than baselines. I think this method is incremental and more rigorous evaluations are needed.

**Reviewer Scores:**

I think all reviewers may keep the scores, as three of them are 6 and one is with 4.

---

### Decision · Program_Chairs · 2026-01-26

Reject